# Effects of body weight-supported Tai Chi Yunshou training on upper limb motor function in stroke patients: A three-arm parallel randomized controlled trial

Liying Zhang[1,2], Xiaoming Yu[1], Wangsheng Liao[3], Jiening Wang[1], Yan Lu[1], Naizhen Wang[3,4‡]*, Zhou Huanxia[1‡]*

1 Department of Rehabilitation, Seventh People's Hospital of Shanghai University of Traditional Chinese Medicine, Shanghai, China, 2 Department of Neurology, Fujian Provincial Governmental Hospital, Fujian, China, 3 Department of Rehabilitation, Fuzhou Second Hospital, Fujian, China, 4 Fujian Medical University School of Health, Fujian, China

‡ NW and HZ contributed equally to this work as correspondent authors.
* 18959115002@189.cn (NW); zhouhuanxia7@163.com (HZ)

## Abstract

### Objectives

To form a unique body weight support-Tai Chi Yunshou (BWS-TCY) training method, apply it to the treatment of upper limb dysfunction after stroke, and provide a new safe and effective treatment method for the clinic.

### Methods

A total of 93 subjects were recruited and randomly divided into conventional rehabilitation treatment (CRT) group, BWS-TCY group and traditional robot-assisted training (RAT) group in equal proportions. Subjects in the CRT group received 60 minutes of CRT daily. Subjects in the BWS-TCY group received 30 minutes of CRT and 30 minutes of BWS-TCY. Subjects in the RAT group received 30 minutes of CRT and 30 minutes of RAT. All interventions were conducted 5 days per week for 12 weeks. Outcome assessments included the Fugl-Meyer Upper Extremity Motor Function Assessment (FMA-UE), Wolf Motor Function Test (WMFT), absolute angular error (AAE), joint range of motion (JMA), modified Barthel Index (MBI), and stroke-related quality of life. Table (SS-QOL). Outcome measures were assessed at baseline, 4 weeks, 8 weeks, and 12 weeks later, and statistical analyzes were performed using two-way repeated measures analysis of variance.

### Results

After 12 weeks of intervention, significant improvements were observed in all evaluation indicators for the three groups of subjects compared to before the intervention. The upper limb motor function (FMA-UE and WMFT) and proprioception (AAE) showed time effects, time × group interaction effects, and group effects. When comparing the groups, the FMA-UE in the BWS-TCY group, as well as the WMFT and MBI, showed statistically significant

**Data Availability Statement:** All relevant data are within the manuscript and its Supporting information files.

**Funding:** This study was founded by "Morning Star" Talent Training Project of Seventh People's Hospital of Shanghai University of Traditional Chinese Medicine (Grant No. QMX2021-08), Youth Research Project of Shanghai Municipal Health Commission (Grant No. 20214Y0285), Fujian Provincial Clinical Medical Research Center for First Aid and Rehabilitation in Orthopedic Trauma (Grant No. 2020Y2014), Medical Discipline Construction Project of Pudong New Area Commission of Health and Family Planning (Grant No. PW2022A-71). The funders had no role in study design, data collection and analysis, decision to publish, or preparation of the manuscript.

**Competing interests:** The authors have declared that no competing interests exist.

**Abbreviations:** AAE, absolute angular error; BWS, body weight support; CRT, conventional rehabilitation therapy; FMA-UE, Fugl-Meyer assessment-Upper Extremity; JMA, joint movement angle; MBI, Modified Barthel Index; RAT, robotic-assisted training; SS-QOL, Stroke-specific quality of life; TCY, Tai Chi Yunshou; WMFT, Wolf Motor Function Test.

differences compared to the CRT group ($P<0.05$), but not statistically significant compared to the RAT group ($P>0.05$). The AAE of the BWS-TCY group showed no statistical difference ($P>0.05$) when compared to the CRT group and RAT group. Furthermore, a time effect was observed on the rotation direction ($P<0.05$), and pairwise comparison between groups revealed that the BWS-TCY group performed better than both the CRT group and the RAT group. After 12 weeks of intervention, there were time effects and interaction effects between BWS-TCY and daily living activities (MBI) and quality of life (SS-QOL), but no group effect was observed. There was no statistical difference between the two groups in SS-QOL. However, there was a statistical difference ($P<0.01$) in MBI between the two groups.

## Conclusions

The 12-week BWS-TCY intervention has been shown to effectively improve upper limb motor function.

## Trial registration

Retrospectively registered at chictr.org.cn on August 31, 2022 [ChiCTR2200063150] https://www.chictr.org.cn/showproj.html?proj=176229.

## 1. Introduction

Stroke is an acute cerebrovascular disease caused by the sudden rupture or occlusion of blood vessels. It is divided into two main types: ischemic stroke, which accounts for 87% of cases, and hemorrhagic stroke, which accounts for 13% [1]. Ischemic stroke occurs when there is ischemia and hypoxia in the brain tissue around the blood vessel supply area, leading to a disruption of local blood supply and resulting in neurological deficits [2]. Hemorrhagic stroke is characterized by spontaneous intracranial hemorrhage, including cerebral hemorrhage and subarachnoid hemorrhage. It is typically caused by arterial rupture due to non-traumatic factors [3, 4]. Stroke is the second most common cause of death and the leading cause of disability worldwide [5, 6]. According to statistics from the World Health Organization, approximately 15 million people worldwide suffer from stroke every year. Out of these, more than 5 million people die from stroke and another 5 million people experience permanent severe disabilities [7]. Moreover, stroke often leads to severe complications in patients, including neuropsychiatric disorders and impairment of motor, sensory, and cognitive abilities [8, 9]. Among long-term stroke survivors, 70%-80% of patients will have various types of functional impairments [10]. This includes 48% with hemiplegia [11], 22% who are unable to walk, and 24%-53% who are partially or completely dependent on daily life activities [12]. Additionally, 55–75% of them still experience upper limb dysfunction 3–6 months after the stroke onset [13]. This dysfunction is characterized by abnormal postures such as scapula withdrawal and sinking, shoulder joint flexion, adduction, internal rotation, elbow joint flexion, forearm supination, finger flexion, etc. Furthermore, 37% of patients have varying degrees of upper limb fine motor impairment [14]. Unilateral upper limb dyskinesia is a frequently observed complication [15, 16]. Individuals experiencing upper limb dysfunction often exhibit restricted joint movement, muscle contraction difficulties, and coordination disorders [17]. Following a stroke, upper limb dysfunction can greatly impede daily activities like eating, dressing, and

washing [18, 19]. This limitation increases patients' reliance on others and negatively impacts their long-term quality of life [20]. Consequently, rehabilitating upper limb dysfunction becomes imperative in enhancing their functional capabilities.

Different rehabilitation techniques and various therapies can be used to restore upper limb function. One such technique is repetitive transcranial magnetic stimulation, a non-invasive brain nerve modulation technology that regulates the cortical excitability of cranial nerves to promote the recovery of upper limb motor function [21, 22]. Another technique, task-oriented bilateral training, focuses on training both the healthy and affected sides together [23, 24]. This helps the affected side imitate the movement pattern of the healthy side, stimulating the memory of corresponding muscles on the affected side and promoting the recovery of motor functions. Virtual reality technology creates a simulation environment that enables virtual interaction in sight, hearing, touch [25]. This technology fully engages patients in training and induces neuroplasticity through repetitive training, enhancing brain movement feedback. Mirror therapy uses visual feedback to compensate for reduced or missing sensory input in the affected upper limb and establish connections between limbs [26, 27]. By converting visual information into active behavior, the mirror neuron system is activated, promoting movement. Robot Assistant Training (RAT) integrates multiple disciplines such as rehabilitation medicine, robotics, situational interaction technology, and control engineering. It is based on the principles of neuroplasticity and motor relearning technology, and offers the advantages of quantification, individualization, and repeatability [28, 29]. Therapies that involve high-intensity repetitive tasks, like RAT, have been found to be highly effective in restoring upper limb function [30]. These therapies offer benefits such as high-intensity repetitive training, good visual feedback, and gravitational support [31]. However, it is important to note that RAT compensates for the affected upper limb through gravity, which can pose challenges to the patient's compliance during the exercise and consequently impact the treatment results [32]. Therefore, body weight support (BWS) may not be the most suitable option for long-term rehabilitation of upper limb dysfunction after a stroke and for improving the mental health of patients. Tai Chi is a traditional Chinese aerobic exercise that involves whole-body movements, including limb wrapping, to help patients regain lost neuromuscular functions [33]. Specifically, Tai Chi Yunshou (TCY) is a low-impact, moderate-intensity exercise that focuses on upper limb movement training. Studies have shown that TCY is effective in improving stroke stability, endurance, coordination, and motor function [34–36]. What sets TCY apart from other exercise interventions is its emphasis on high coordination of the upper limbs, complex motor control, and hand-eye coordination. This type of exercise activates the cerebral cortex and brain regions to a greater extent, leading to long-term enhancement or remodeling of functional connections in the brain [36]. However, completing TCY exercises requires better motor functions, such as muscle strength (Lovett >2) and joint range of motion [37]. Additionally, TCY exercise is generally suitable for patients in the late stage of stroke (Brunnstrom stage >3) [35]. Unfortunately, these abilities are nearly impossible for most early-stage stroke patients. Therefore, it is crucial to develop interventions that are simple and not limited by functional impairment. These interventions should be included in current stroke rehabilitation programs to enable patients to persist in training and obtain sustained benefits from treatment.

The effectiveness of TCY in the late stage of stroke for upper limb rehabilitation has been confirmed by previous studies [35, 38]. However, according to the guidelines of the American Stroke Association, early rehabilitation intervention leads to better outcomes [39]. Therefore, our research focuses on exploring the use of TCY in the early stage of stroke. Previous studies have demonstrated that weight-supported Tai Chi footwork training, which employs a suspension device within a balance bar, enhances lower limb motor function and balance in patients

with early-stage stroke [40, 41]. By integrating Tai Chi with suspension devices, patients who are unable to fully bear weight can commence training promptly. However, this training method necessitates the presence of at least two therapists simultaneously, leading to increased time and labor demands. In contrast, rehabilitation robots provide gravity compensation and are more user-friendly [42]. Rehabilitation robotic exoskeletons and joysticks not only provide gravity compensation but also offer power-assisted training, resistance training, and passive training. The exoskeleton provides maximum flexibility in terms of weight support and control strategies, making it easily adaptable and usable in clinical settings.

Based on the aforementioned benefits, we have developed a program utilizing a rehabilitation robot to facilitate the completion of TCY movements by driving the affected upper limb with the robot's mechanical arm. Therefore, BWS-TCY may be a suitable exercise for enhancing upper limb function in stroke patients due to its unique exercise method. Firstly, participants are required to maintain stability in their upper limbs during TCY in order to ensure smooth movements, making TCY an effective way to stimulate muscle contraction in the upper limb. Secondly, patients face challenges in controlling their speed and frequently adjusting incline angles during training. By performing TCY, coordinated movements can be trained and the flexibility of upper limb joints can be improved. Moreover, patients are required to recall and reproduce these movements during training, which can enhance their cognitive abilities, hand-eye coordination, and sense of realism. Lastly, the robot system provides vivid and engaging animations as visual feedback to enhance patient motivation during training. Therefore, TCY holds great potential in promoting the recovery of upper limb function and improving the mental well-being of stroke patients. The use of BWS-TCY can facilitate better and faster learning of TCY movements, enabling stroke survivors to independently practice at home after discharge without the need for professional guidance or supervision. Enhancing the ability of stroke survivors to carry out their own rehabilitation plans anytime and anywhere should also be considered an important rehabilitation goal.

We proposed a novel intervention using BWS-TCY to test whether it has a better rehabilitation effect than RAT in stroke patients. This study included three groups: conventional rehabilitation treatment (CRT) group, CRT+BWS-TCY group, and CRT+ RAT group. Based on the pilot study involving 15 patients, we calculated the mean and standard deviation of the scores for each evaluation index both before and after the intervention. We hypothesized that the three groups would show differential improvements in upper limb motor function, motor control, and joint range of motion, and that the rehabilitation effect of BWS-TCY would surpass that of both RAT and CRT. Furthermore, we posited that the rehabilitation effect of RAT would be greater than that of CRT. The findings of this study will contribute to optimizing the existing rehabilitation treatment process by integrating rehabilitation intervention, aiming to intervene in rehabilitation treatment earlier and more effectively. Additionally, it will provide stroke patients with a reference for choosing rehabilitation training programs.

## 2. Study methods

### 2.1 Study design

This study was conducted as a single-center, three-arm, parallel-group, rater-blinded randomized controlled trial (see Fig 1). From August 2022 to March 2023, the participants were recruited through the Department of Rehabilitation Medicine and Department of Neurology of Shanghai Seventh People's Hospital, promotional posters, and therapist recommendations. Prior to subject recruitment, all patients were provided with information about the study.

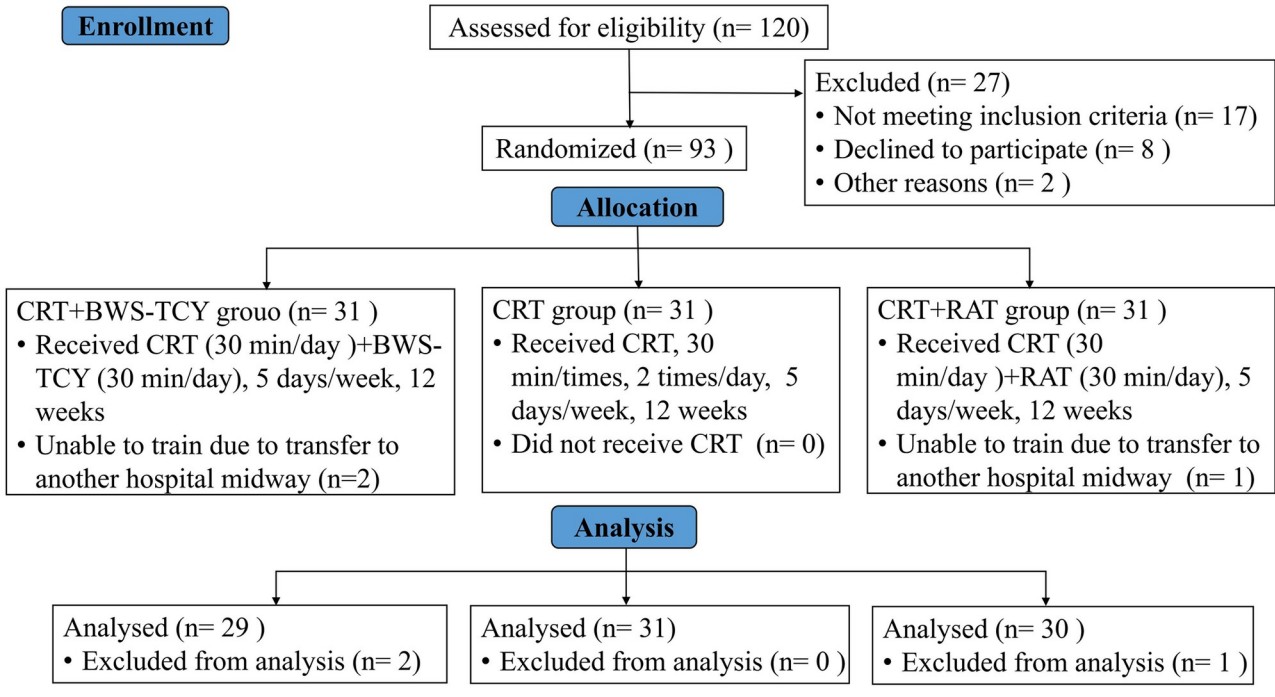

**Fig 1. CONSORT 2010 flow diagram.**

## 2.2 Eligibility criteria

### 2.2.1 Inclusion criteria.

1. Presence of stroke documented by CT or MRI, including ischemic and hemorrhagic stroke;

2. Subacute stage of stroke (first onset, duration of disease within 6 months);

3. Ability to sit and balance without upper limb support, and able to tolerate at least half an hour of training or testing;

4. Brunnstrom classification of the patient's upper limb is ≤4;

5. Stable blood pressure (less than 160/100 mmHg);

6. Good cognitive ability (Mini-Mental State Examinatione score ≥22) [43];

7. Good muscle tone (modified Ashworth classification < level 2);

8. No severe visual impairment or visual field defect;

9. Age between 35–80 years old, with no gender limit.

### 2.2.2 Exclusion criteria.

1. Presence of severe bone and joint disease, muscle disease, or other neurological diseases, or a history of upper limb surgery;

2. History of skull fracture and/or severe head injury; presence of aphasia, hearing impairment, or other communication difficulties that hinder normal communication with others;

3. Manifestation of obvious shoulder pain, with a pain score at rest exceeding 5 [44];

4. Duration of the disease exceeding 6 months since onset;

5. Severe visual or hearing impairment that hampers the ability to cooperate with training;

6. Coexistence of severe liver, kidney, heart, lung, and blood disorders, as well as other systemic diseases;

7. Participation in other ongoing clinical studies.

**2.2.3 Elimination and discontinuation criteria.**

1. Patients who provided incomplete information, leading to missing data on the main outcome indicators;

2. Patients who provided false information;

3. Patients with other systemic diseases that made them unsuitable to continue participating in the study;

4. Patients who experienced sudden illness or adverse events during the trial intervention period and were unable to continue or participate in the training;

5. Participants who voluntarily withdrew from the study;

6. Participants who experienced adverse events.

## 2.3 Ethics approval and consent to participate

All study procedures were conducted in accordance with the current version of the Declaration of Helsinki (see www.wma.net for details). This study has received approval from the Medical Ethics Committee of Shanghai Seventh People's Hospital (No. 2022-7th-HIRB-022). The Chinese clinical trial registration number is ChiCTR 2200063150. Prior to entering the study, participants were thoroughly informed about the study, potential risks, and other relevant matters. They provided their informed consent by signing a consent form before randomization.

## 2.4 Sample size

We utilized G*Power software (v3.1.9.2, University Dusseldorf, Germany; available for download from http://www.psychologie.hhu.de) to determine the sample size for our study. The primary outcome evaluation index considered in this study was FMA-UE. Prior to commencing the formal experiment, 15 patients who met the inclusion criteria were divided into groups in a 1:1:1 ratio. Sample size calculation was conducted for the main objective, which focused on the treatment effect 4 weeks after intervention. The results of the preliminary experiment revealed the following mean ± standard deviation values after 4 weeks of intervention: TCY +CRT treatment (35.5±9.5), RAT+CRT treatment (34.28±9.58), and CRT alone (29.00±10.2). Based on the G*power two-factor repeated measures analysis of variance (ANOVA) F test, the effect size was calculated to be 0.2765, derived from the mean and standard deviation of the FMA-UE scores across the three patient groups. A total sample size of 84 cases was determined for a two-tailed test with a power of 80%, repeat measurement four times, with a correlation among repeated measures of 5%, and a significance level of 5% (alpha error). Considering a 1:1:1 allocation ratio and accounting for a 10% dropout rate, we estimated that a final total of 93 patients (31 in each group) would be required.

## 2.5 Randomized grouping and allocation concealment

Patients were assigned treatment using a randomization procedure where the randomization procedure was conducted through a software that utilized random permuted blocks. Patients were assigned to their treatment arm according to stratified random lists that were balanced in blocks of various sizes in random sequence. To conceal allocation sequence, random permuted blocks with sizes 2 and 4 were used. The subjects, totaling 93, were evenly distributed into three groups: BWS-TCY group, CRT group, and RAT group, each consisting of 31 cases. The random sequence was created by an independent professional statistician using SPSS software (IBM Corp., IBM SPSS Statistics, V25, Armonk, NY, USA), with the random number seed set to 20210608. After baseline testing, each participant received an envelope containing a randomly assigned serial number to determine their group. Throughout this process, the statisticians, outcome assessors, and data analysts were blinded to the study's recruitment, intervention, and evaluation. The randomization assignments were placed in sealed opaque envelopes by an independent researcher who was also blinded to the trial. The identity of each participant is represented by a serial number, which is numbered according to their entry into the group. The letters 'A', 'B', and 'C' are used to represent the assigned groups, with 'A' corresponding to the BWS-TCY group, 'B' to the CRT group, and 'C' to the RAT group. The randomization process was conducted by an independent researcher. Three rehabilitation therapists were responsible for recruiting and assigning participants to conduct the intervention. Due to significant differences between the groups, neither the researchers nor the therapists were blinded. The study was designated as open-label, so the patients were not blinded.

## 2.6 Upper limb rehabilitation robot equipment and BWS-TCY program design

The upper limb rehabilitation robot used for BWS-TCY training and RAT training is Fourier Intelligence Co., Ltd.'s product called Fourier Arm Motus EMU (shown in Fig 2). It is a three-dimensional upper limb rehabilitation robot that operates on terminal control. The robot offers real-time, dynamic compensation for gravity and includes various games for upper limb training (see Fig 3). These games involve activities like stretching the affected side's upper limb to musical notes in different directions (referred to as the 'grabbing the musical notes' game), simulating fishing actions and putting fish into boxes of different colors (known as the 'fishing game'), online imitation of table tennis with teammates (known as the 'table tennis game'), and activity training for various joints of the upper limbs.

BWS-TCY training utilizes the EMU robot's robotic arm to guide the affected upper limb in completing TCY movements. To facilitate this, a modified set of procedures was developed (see Fig 4). The program consists of two modules: the first module focuses on generating customized TCY motion trajectories based on the specific characteristics of each patient, which are then saved for future training sessions. The second module is the training module, where the saved TCY trajectory can be selected and different modes (passive, assisted, and resistance) can be chosen based on the severity of upper limb impairment. The system also offers adjustable gravity compensation, as well as customizable time and range of motion. During training, the system computer simultaneously plays TCY action videos and soothing music.

## 2.7 Interventions

All patients received rehabilitation intervention in addition to routine medical treatment and daily care in the hospital. The patient's safety status was continuously recorded throughout the

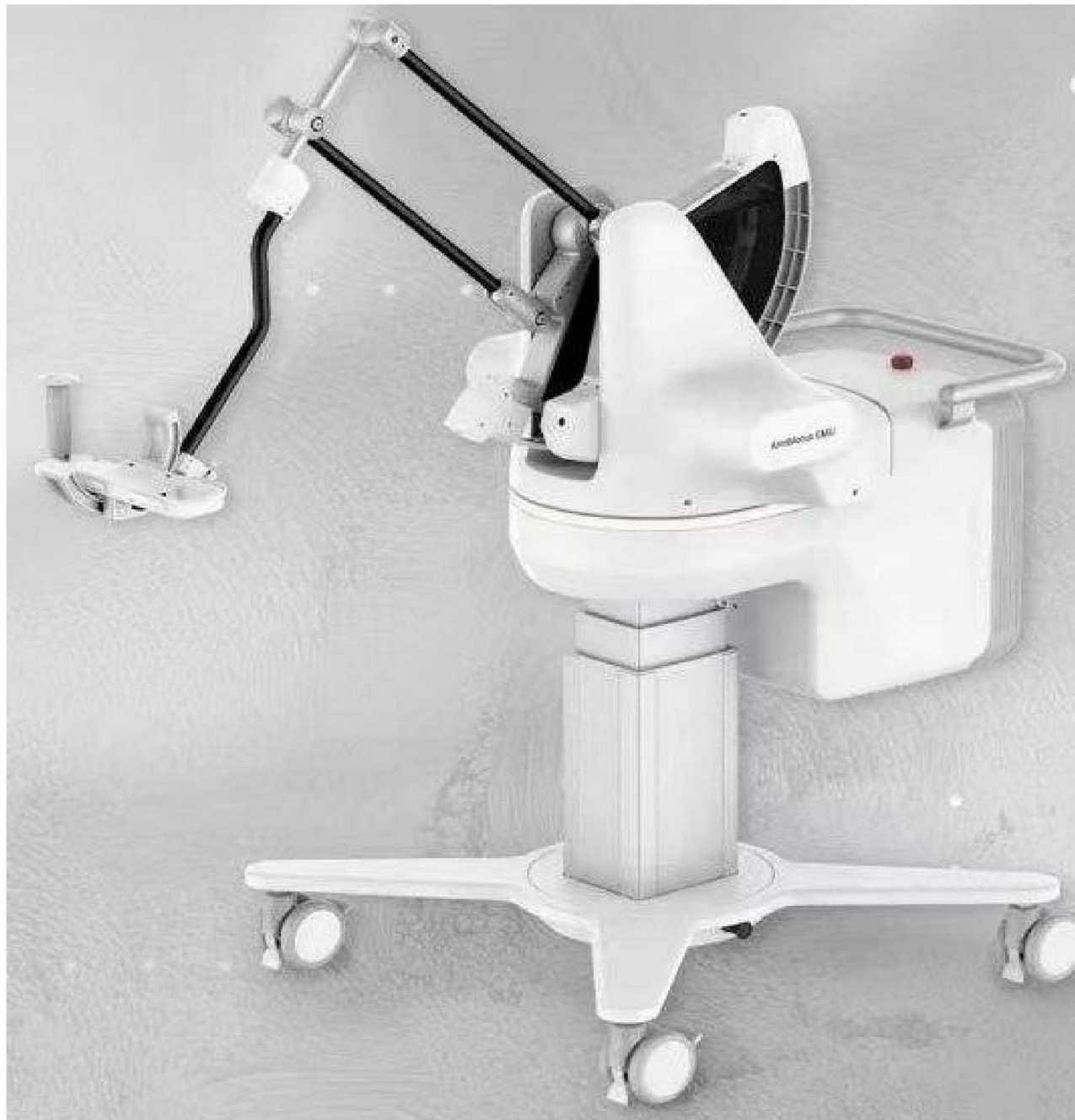

**Fig 2. Rehabilitation robot Fourier Arm Motus EMU.**

intervention. The rehabilitation intervention lasted for 12 weeks, with sessions of 60 minutes per day, 5 days per week.

**2.7.1 Intervention plan for CRT group.** In the CRT group, patients received 60 minutes of CRT treatment every day, divided into two 30-minute sessions in the morning and afternoon. The training primarily focuses on the affected side and involves physical therapy, occupational therapy, and rehabilitation care. The training program includes various

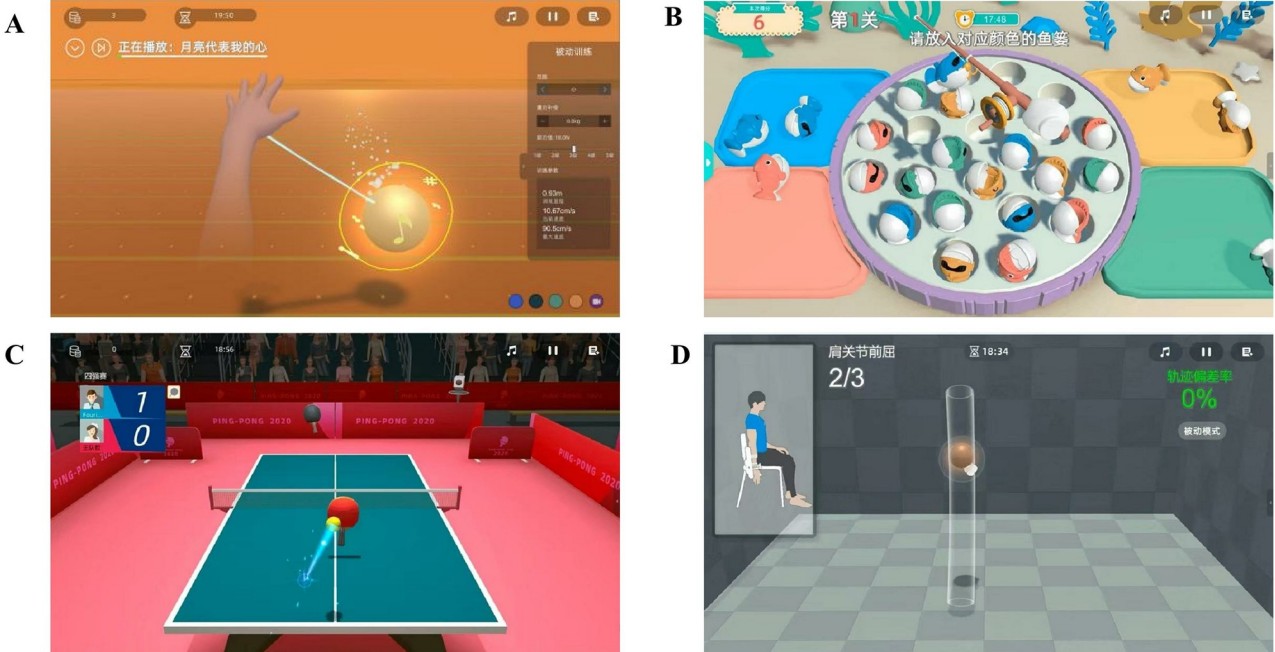

**Fig 3. Upper limb training games provided by rehabilitation robot Fourier Arm Motus EMU.** (**A**) Notes catching game; (**B**) Fishing games; (**C**) Table tennis game; (**D**) Upper limb joint range of motion training.

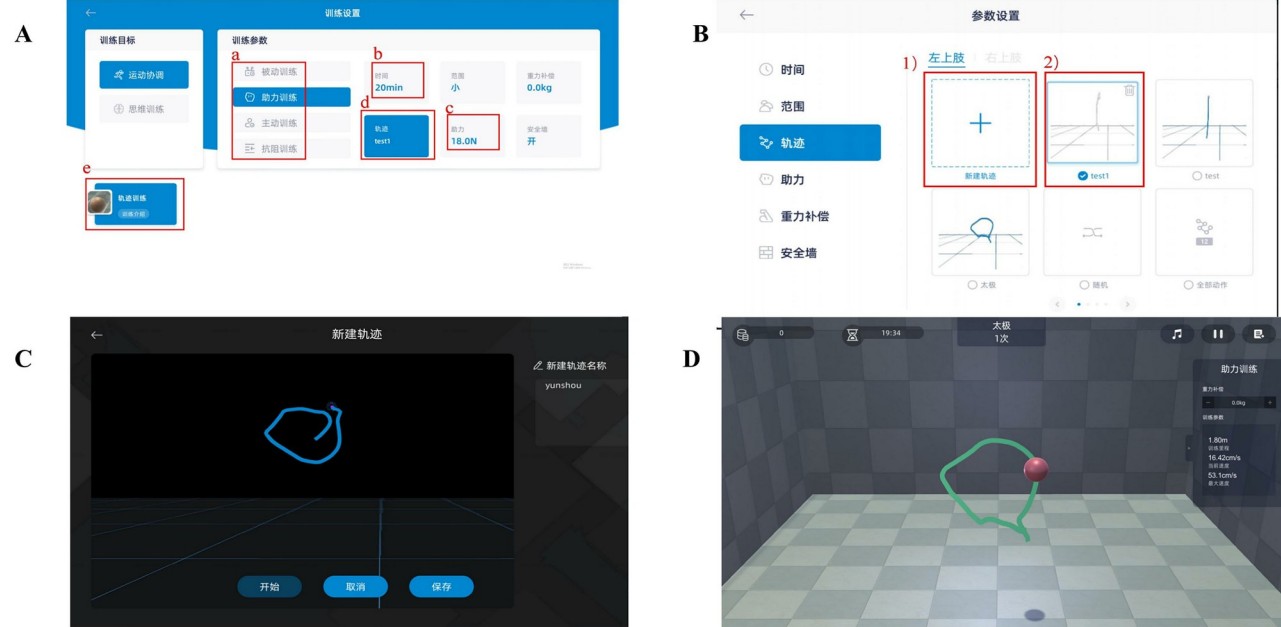

**Fig 4. Systerm program of BWS-TCY.** (**A**) Training parameter; (**B**) Motion path selection; (**C**)Motion trajectory rendering; (**D**)Follow the existing motion path training. Note: a.Selecting training mode; b.Determining training duration; c.Choosing power value; d.Choosing the trajectory of motion; 1)Ploting a new trajectory; 2)Choosing the established trajectory.

activities such as passive joint exercises, weight training with the assistance of the healthy hand, anti-spasm model training, functional activity series training for shoulder, elbow, and wrist joints, finger function and fine movement rehabilitation training, proper limb positioning, daily living ability training, turning over in bed training, balance training, and therapies such as proprioceptive neuromuscular stimulation technology therapy and Rood therapy.

**2.7.2 Intervention plan for RAT+CRT group.** After 30 minutes of CRT, the RAT training session began. The patient is seated in front of a computer and next to a robot with a mechanical arm. The height of the robotic arm is adjusted to be at the same level as the patient's shoulders. The patient is instructed to relax naturally and look straight at the computer screen in front of them. The affected forearm of the patient is secured to the handle of the robotic arm. The therapist then selects upper limb games based on the patient's preferences and recommendations. Before entering the game interface, the therapist adjusts parameters such as gravity compensation value, training mode (active, assisted, or passive), training time, and music. In the note grabbing game, the patient is required to touch notes from different directions using the affected upper limb and count the number of touches. In the fishing game, the robot arm simulates a real fishing scene and places fish hooks on small fish of different colors that constantly change positions. The small fish are then placed in corresponding color fish frames on both sides. In the table tennis game, the patient can compete online with patients from different locations, imitate real movements, hit the ball back from the opponent, and the competition is recorded. For upper limb joint activity training, the patient can initially passively complete the movements following the system's instructions (this step can be skipped if already proficient). Once proficient, the patient can select a training mode based on the condition of their upper limbs. The joints trained include 8 movements of the shoulder joint (forward flexion, posterior extension, abduction, adduction, external rotation, internal rotation, horizontal abduction, horizontal adduction), as well as elbow flexion and extension, and forearm pronation and supination. Fig 5 shows the patient training scenario.

**2.7.3 Intervention plan for BWS-TCY+CRT group.** In the CRT+BWS-TCY group, patients underwent 30 minutes of CRT treatment and 30 minutes of BWS-TCY training daily. During BWS-TCY training, patients watched videos and learned TCY movements with the assistance of professionally trained therapists. They were required to skillfully apply these movements. The exoskeleton rocker of the rehabilitation robot, which can customize the TCY movement trajectory, helped the patients with restricted limbs to complete TCY movements. The training process involved the following steps: ① Sitting upright, relaxing the body naturally, keeping the head upright, and maintaining direct eye contact with the person on the screen in front. This top-down conscious guidance facilitated overall body relaxation. ② The affected forearm was fixed on the handle, with the hand in an extended position. ③ The patient followed the pre-saved TCY motion trajectory, allowing the robot's mechanical arm to assist in carrying out the TCY action. This involved moving the shoulders, elbows, and wrists together to draw a circle clockwise from top to bottom and from inside to outside. Studies have demonstrated that as the weight support ratio increases, the degree of muscle activation gradually decreases [40]. Therefore, the weight support ratio was set at 40% at the beginning of the 12-week intervention program. The program followed the principle of starting from easy to difficult and repeating intensively. It was divided into 5 different weight support stages: Weeks 1–3: 40%; Weeks 4–7: 30%; Weeks 8–10: 20%; Weeks 11–12: 0%. Fig 6 illustrates the typical patient practice diagrams for BWS-TCY training.

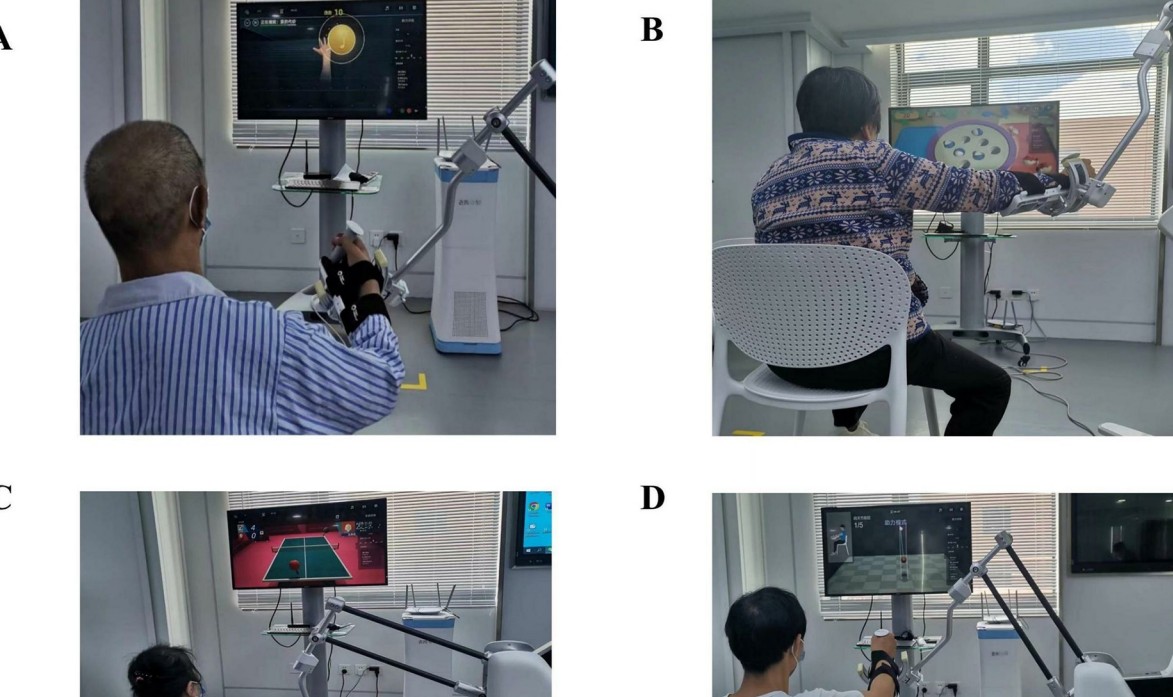

**Fig 5. Rehabilitation robot upper limb training scene.** (**A**) Notes catching game scene; (**B**) Fishing games scene; (**C**) Table tennis game scene; (**D**) Upper limb joint range of motion training scene.

## 2.8 Outcome measures

Demographic characteristics were obtained from questionnaires completed by assessors during recruitment and from patient records in the hospital management system. All outcome measures were assessed at baseline, after 4 weeks and 8 weeks of intervention, and finally after 12 weeks of intervention.

**2.8.1 Fugl-Meyer motor assessment- upper extremity.** The Fugl-Meyer motor assessment- upper extremity (FMA-UE) is the main index used in this study to measure upper limb motor dysfunction after stroke. It is a cost-effective clinical examination method that is widely used in stroke patients due to its reasonable design, simplicity, and ease of use [45]. The FMA-UE assesses reflex activities, shoulder, elbow, and wrist joint movement, as well as coordination. It consists of 8 aspects and 33 items, with each item scored on a scale of 0 to 2 points. The total score ranges from 0 to 66 points.

**2.8.2 Wolf Motor Function Test.** The Wolf Motor Function Test (WMFT) is a rehabilitation scale used to assess the motor function of the upper limbs in individuals who have had a stroke [46]. Unlike the FMA, which primarily assesses the coordination function of stroke patients, the WMFT can evaluate both impairment and the impact of training on disability

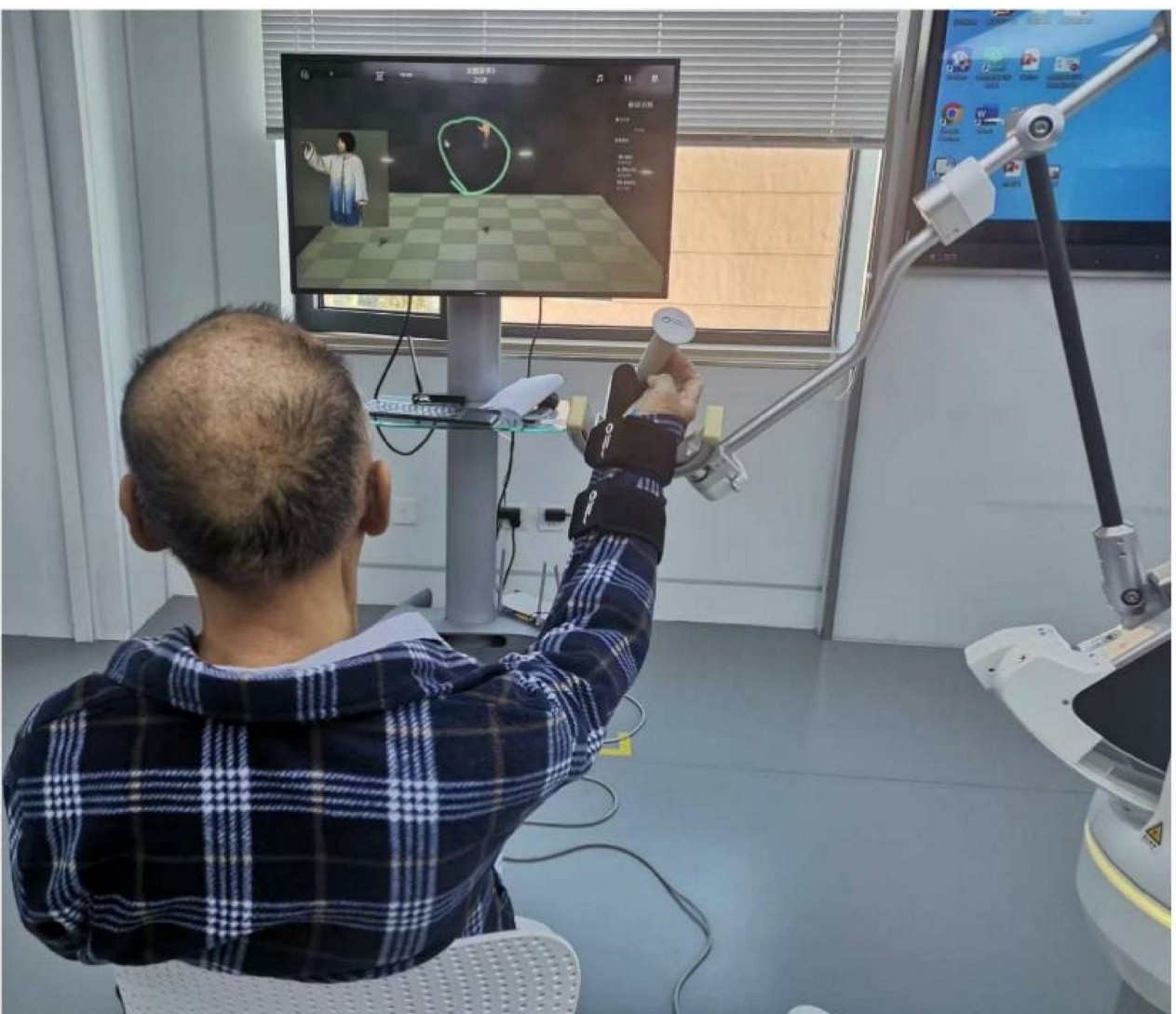

**Fig 6. BWS-TCY training scenario.**

[47]. Additionally, it can reflect the effects of various functional task trainings on patients. The test consists of 15 items, with the first six focusing on simple joint movements and the remaining nine involving compound functional movements. Each action is timed and scored based on the quality of the movement using a six-grade scale ranging from 0 to 5 points.

**2.8.3 Absolute angular error.** Absolute angular error (AAE) is defined as the absolute difference between the target angle and the patient's perceived elbow flexion angle. It is used to assess the patient's proprioception and motor control [36]. In this study, we set the target angle to 90° of elbow flexion and an angular velocity of 2°/s, as measured by the upper limb intelligent feedback training system for evaluating upper limb proprioception. During the training session, the patient wears eye masks and earmuffs to minimize any external influence on proprioception. The patient is instructed to move the affected upper limb from the vertical position to the target position, memorize the target position, take a 10-second rest, and then repeat the same movement. The difference between the target position and the measured

position is recorded as the absolute error angle. A smaller absolute error angle indicates a better position sense.

**2.8.4 Joint Movement Angle.**   Joint Movement Angle (JMA) is a method that utilizes a goniometer to measure the maximum angle of active movement in the joints of the upper limb. The measured upper limb joint movements include shoulder flexion, shoulder extension, shoulder abduction, shoulder adduction, shoulder external rotation, shoulder internal rotation, elbow flexion, forearm pronation, and forearm supination. It is important to note that a larger angle of active movement indicates better motor function.

**2.8.5 Modified Barthel Index.**   The improvement in basic activities of daily living can be assessed using the Modified Barthel Index (MBI), a commonly used tool for evaluating the ability of stroke patients to perform daily tasks. The MBI consists of 10 tasks, which are scored based on the time and assistance required by the patient to complete them. These tasks include eating, bathing, dressing, washing and grooming, control of defecation, control of urination, using the toilet, going up and down stairs, transferring from bed to chair, and walking 45 meters on level ground. Transferring from bed to chair and walking 45 meters on level ground have the highest scores of 15 points, while grooming and bathing have a maximum score of 5 points. The other six tasks have a maximum score of 10 points each. The lowest score for each task is 0 points, and the total score ranges from 0 to 100. A lower score indicates a greater dependence on care, while a score of 60 or higher suggests the ability to take care of oneself.

**2.8.6 Stroke-specific quality of life.**   Stroke-specific quality of life (SS-QOL) is a patient-reported prognostic indicator used to assess the health-related quality of life in stroke patients. It can also be reported as a secondary outcome. The content of SS-QOL includes 49 items distributed across 12 domains, such as energy, family roles, language, mobility, mood, personality, self-care, social roles, thinking, upper body function, vision, and work/productivity. Each domain is scored separately, with a maximum score of 5 points for each item. The scores are then calculated to obtain the total score. A higher score indicates better functioning.

## 2.9 Security assessment

Prior to enrollment, patients are required to undergo evaluations for muscle strength, muscle tone, Brunnstrom stage, and Mini-Mental State Examinatione. Additionally, a general physical examination including respiration, heart rate, blood pressure, pulse, body temperature, etc. is conducted. In light of the global epidemic of the new coronavirus, all patients must undergo a nucleic acid test, with the results being accurately documented and the therapist being informed to take necessary protective measures. Any adverse events that occur during the study period are recorded on the case report form. Adverse events refer to any unfortunate medical events, such as cardiovascular events, cerebrovascular events, and falls, that take place throughout the study.

## 2.10 Data processing and analysis

**2.10.1 data processing.**   The research reports of eligible individuals will be completed by the evaluators of this project team. These reports will include an informed consent form, research flow chart, general situation questionnaire, evaluation scale, balance assessment results registration form, and other relevant documents. The completed research reports will undergo a thorough review by specialized project managers. Additionally, specialized data entry personnel will use Excel to create a database for data entry.

**2.10.2 Statistical analysis.**   Statistical analysis of the data was conducted using SPSS 25.0. The count data and measurement data of baseline characteristics were compared using the Chi-square test and one-way ANOVA, respectively. A two-way repeated ANOVA was

employed to examine the interaction and main effects of the intervention method and assessment time. The impact of the intervention will be analyzed by comparing changes in affected upper limb function between groups using analysis of covariance on change scores, with baseline as a covariate and adjustment for potential confounders. If significant differences are found, post hoc analysis will be conducted using Tukey's test. Descriptive statistics for each group will be used to describe demographic characteristics and other baseline values. 0.05 was used as the test level, and the Holm's adjustment P value was used as the basis for significance. As a sensitivity analysis, an intention-to-treat analysis, which involves analyzing participants in the assigned treatment group, can be performed using only the observed data.

## 3. Results

### 3.1 Patients' completion of the trial

In this study, 120 stroke patients were recruited from the Department of Neurology and Rehabilitation Medicine of Shanghai Seventh People's Hospital between August 2022 and March 2023. Out of the recruited patients, 93 met the inclusion criteria and agreed to participate. They were then randomly divided into three groups: BWS-TCY+CRT group, CRT group, and RAT+CRT group, with 31 cases in each group. Throughout the 12-week intervention period, only 3 patients dropped out due to transfer, specifically 2 from the BWS-TCY+CRT group and 1 from the RAT+CRT group. Therefore, a total of 90 stroke patients were included in the analysis.

### 3.2 Results of baseline comparison of three groups

According to Table 1, the baseline characteristics of the three groups of patients were comparable.

### 3.3 Results of upper limb motor function

The upper limb motor functions of the three groups of participants were evaluated using the Fugl-Meyer Assessment for Upper Extremity (FMA-UE) and the Wolf Motor Function Test (WMFT) indicators. The evaluations were conducted before the intervention, after 4 weeks, after 8 weeks, and after 12 weeks of intervention. The data for both indicators followed a normal distribution, and the Mauchly sphericity test was performed (refer to Table 2). However, the assumption of sphericity was not met ($P<0.001$), so the Greenhouse-Greisser correction was applied. Table 3 shows that there were significant time effects, time × group interaction effects, and group effects in the FMA-UE and WMFT scores of the three groups ($P<0.01$). By referring to Table 4, it can be observed that the upper limb motor functions (FMA-UE and WMFT) of stroke patients continue to improve over time. The differences between the scale scores after 4 weeks, 8 weeks, and 12 weeks of intervention and the scores before the intervention are presented in Table 5. It can be noted that these differences gradually increase as time progresses. The Shapiro-Wilk test and Levine's equality of variance test were conducted to assess the normal distribution and homogeneity of variances. The results indicated that only the difference between the scores after 12 weeks of intervention and before the intervention met the requirements ($P>0.05$). Therefore, a two independent samples t-test was used to compare the FMA-UE and WMFT among the three groups of patients (refer to Table 6). The findings are as follows: ① A significant difference was observed between the BWS-TCY group and the CRT group ($P<0.001$). The BWS-TCY group had higher scores in FMA-UE and WMFT compared to the CRT group. ② There was no statistical difference in FMA-UE between the

**Table 1. Comparison of baseline characteristics of the three groups of patients.**

| Characteristics | CRT+BWS-TCY (n = 29) | CRT (n = 31) | CRT+RAT (n = 30) |
|---|---|---|---|
| Age, mean±SD | 59.41±11.78 | 63.13±12.51 | 65.44±13.36 |
| Gender (male, %) | 23 (67.6%) | 25 (73.5%) | 20 (58.8%) |
| Duration of disease (days) | 62.83±29.85 | 63.23±32.58 | 65.00±24.46 |
| Type of disease (Ischemic, %) | 21 (72.4%) | 21 (67.7%) | 22 (73.3%) |
| Hemiplegic side (Lelf, %) | 8 (27.6%) | 13 (41.9%) | 5 (16.7%) |
| Handedness (Right, %) | 27 (93.1%) | 29 (93.5%) | 28 (93.3%) |
| Heart rate (beats/minute) | 80.10±8.17 | 75.19±6.87 | 79.17±9.51 |
| Systolic blood pressure (mmHg) | 139.45±12.4 | 136.77±16.8 | 134.73±16.6 |
| Diastolic blood pressure (mmHg) | 84.45±7.49 | 79.39±9.59 | 81.60±9.41 |
| NIHSS, mean±SD | 15.66±3.55 | 16.26±3.10 | 15.17±3.14 |
| Brunnstrom staging | | | |
| Phase I | 5 (17.2%) | 3(9.7%) | 2 (6.7%) |
| Phase II | 10 (34.5%) | 11 (35.5%) | 10 (33.3%) |
| Phase III | 11 (37.9%) | 15 (48.4%) | 12 (40.0%) |
| Phase IV | 3 (10.3%) | 2 (6.5%) | 6 (20.0%) |
| Upper limb manual muscle strength test | | | |
| Level 0 | 0 (0%) | 1 (3.2%) | 0 (0%) |
| Level 1 | 4 (13.8%) | 6 (19.4%) | 3 (10.0%) |
| Level 2 | 13 (44.8%) | 18 (58.1%) | 21 (70.0%) |
| Level 3 | 12 (41.4%) | 6 (19.4%) | 6 (20.0%) |
| FMA-UE (0–66, score) | 25.28±9.27 | 25.09±9.56 | 27.73±10.76 |
| WMFT (0–75, score) | 29.17±8.68 | 29.13±6.99 | 27.97±7.52 |
| AEE (0–90, ˚) | 14.87±4.59 | 15.61±4.76 | 15.88±5.02 |
| MBI (0–100, score) | 46.38±14.01 | 47.74±9.99 | 48.33±10.61 |
| SS-QOL(49–245, score) | 99.79±13.81 | 99.58±12.39 | 99.37±13.02 |
| Upper limb joint movement angle | | | |
| Shoulder flexion | 45±20.92 | 51.50±25.18 | 49.35±22.89 |
| Shoulder extension | 17.93±9.28 | 19.58±8.99 | 19.33±9.66 |
| Shoulder abduction | 37.14±30.75 | 33.42±25.93 | 40.37±34.44 |
| Shoulder adduction | 14.21±7.98 | 13.16±7.01 | 15.23±9.74 |
| Shoulder external rotation | 20.97±12.18 | 18.00±11.51 | 22.40±13.38 |
| Shoulder internal rotation | 24.52±13.15 | 21.32±12.29 | 26.13±14.51 |
| Elbow flexion | 28.97±29.49 | 27.26±24.73 | 33.00±32.85 |
| Forearm pronation | 28.59±4.09 | 25.68±12.96 | 30.03±14.35 |
| Forearm supination | 20.62±12.85 | 18.42±11.46 | 21.93±13.38 |

BWS-TCY group and the RAT group ($P$>0.05), but a significant difference was found in WMFT ($P$<0.05). The RAT group had higher scores in WMFT compared to the BWS-TCY group. ③ Compared to the CRT group, the RAT group showed a significant statistical difference ($P$<0.01) and had higher scores than the CRT group.

**Table 2. Repeated measurement of Mauchly spherical test for upper limb motor function.**

| Outcome measures | Mauchly W value | Approximate chi-square value | Significance | Degrees of freedom | Greenhouse-Greisser correction |
|---|---|---|---|---|---|
| FMA-UE | 0.160 | 157.208 | <0.001 | 5 | 0.496 |
| WMFT | 0.236 | 123.722 | <0.001 | 5 | 0.537 |

**Table 3. Repeated measures analysis of variance for upper limb motor function.**

| Outcome measures | Scourses | Sum of Squares | Degrees of freedom | Mean Square | *F* value | Significance | Eta Squared |
|---|---|---|---|---|---|---|---|
| FMA-UE | Time | 19020.42 | 1.487 | 7499.84 | 285.87 | <0.001 | 0.77 |
| | Group×Time | 1279.92 | 2.957 | 357.57 | 9.62 | <0.001 | 0.18 |
| | Group | 1955.29 | 2 | 1313.66 | 3.67 | 0.030 | 0.08 |
| WMFT | Time | 18916.43 | 1.612 | 11735.78 | 392.52 | <0.001 | 0.82 |
| | Group×Time | 839.88 | 3.224 | 260.53 | 8.71 | <0.001 | 0.17 |
| | Group | 1878.65 | 2 | 939.33 | 4.25 | 0.017 | 0.09 |

**Table 4. Ratings of upper limb motor function in three groups at different time points.**

| Outcome measures | Groups | Time point | Mean | Standard deviation | Confidence Interval (%) Upper | Lower |
|---|---|---|---|---|---|---|
| FMA-UE | CRT+BWS-TCY | before intervention | 25.28 | 9.27 | 21.75 | 28.80 |
| | | 4 weeks after intervention | 35.62 | 8.70 | 32.31 | 38.93 |
| | | 8 weeks after intervention | 42.48 | 8.70 | 39.17 | 45.79 |
| | | 12 weeks after intervention | 52.45 | 8.91 | 49.06 | 55.84 |
| | CRT | before intervention | 25.10 | 9.56 | 32.31 | 38.93 |
| | | 4 weeks after intervention | 32.16 | 8.69 | 28.97 | 35.35 |
| | | 8 weeks after intervention | 35.26 | 8.82 | 32.02 | 38.49 |
| | | 12 weeks after intervention | 40.65 | 10.27 | 36.88 | 44.41 |
| | CRT+RAT | before intervention | 27.73 | 10.76 | 23.71 | 31.75 |
| | | 4 weeks after intervention | 34.83 | 8.34 | 31.72 | 37.95 |
| | | 8 weeks after intervention | 39.00 | 8.43 | 35.85 | 42.15 |
| | | 12 weeks after intervention | 48.93 | 6.51 | 46.50 | 51.36 |
| WMFT | CRT+BWS-TCY | before intervention | 29.17 | 8.68 | 25.93 | 32.41 |
| | | 4 weeks after intervention | 34.73 | 8.56 | 28.59 | 33.61 |
| | | 8 weeks after intervention | 43.20 | 9.49 | 32.36 | 37.64 |
| | | 12 weeks after intervention | 49.10 | 8.58 | 45.89 | 52.31 |
| | CRT | before intervention | 29.13 | 6.99 | 25.93 | 32.41 |
| | | 4 weeks after intervention | 31.10 | 6.73 | 28.59 | 33.61 |
| | | 8 weeks after intervention | 35.00 | 7.07 | 32.36 | 37.64 |
| | | 12 weeks after intervention | 38.43 | 7.93 | 35.47 | 41.39 |
| | CRT+RAT | before intervention | 27.97 | 7.52 | 25.16 | 30.78 |
| | | 4 weeks after intervention | 33.17 | 7.19 | 30.48 | 35.85 |
| | | 8 weeks after intervention | 43.00 | 6.06 | 40.74 | 45.26 |
| | | 12 weeks after intervention | 48.97 | 5.57 | 46.89 | 51.05 |

**Table 5. Differences in upper limb motor function between the three groups after the intervention and before the intervention (Mean±SD, score).**

| Outcome measures | Groups | Difference between 4 weeks after intervention and before intervention | Difference between 8 weeks after intervention and before intervention | Difference between 12 weeks after intervention and before intervention |
|---|---|---|---|---|
| FMA-UE | CRT+BWS-TCY | 10.34±8.84 | 17.21±9.17 | 27.17±9.64 |
| | CRT | 7.06±8.33 | 10.16±8.71 | 15.55±9.04 |
| | CRT+RAT | 8.13±8.3 | 11.27±7.29 | 17.30±9.26 |
| WMFT | CRT+BWS-TCY | 8.59±7.41 | 15.69±7.87 | 24.45±6.89 |
| | CRT | 6.29±7.99 | 9.32±8.42 | 14.81±8.73 |
| | CRT+RAT | 7.03±7.00 | 12.37±5.28 | 20.30±6.57 |

**Table 6. Comparison of upper limb motor function scores among the three groups after 12 weeks of intervention.**

| Outcome measures | Comparison between groups | t value | P value (two-tailed) | Mean Difference | Standard error value |
|---|---|---|---|---|---|
| FMA-UE | CRT+BWS-TCY *VS* CRT | 4.740 | <0.001 | 11.80 | 2.49 |
| | CRT+BWS-TCY *VS* CRT+RAT | 1.802 | 0.077 | 4.39 | 2.44 |
| | CRT+RAT *VS* CRT | 3.242 | 0.002 | 7.41 | 2.89 |
| WMFT | CRT+BWS-TCY VS CRT | 4.720 | <0.001 | 9.64 | 2.02 |
| | CRT+BWS-TCY VS CRT+RAT | 2.365 | 0.021 | 4.15 | 1.75 |
| | CRT+RAT VS CRT | 2.770 | 0.007 | 5.49 | 1.97 |

## 3.4 Results of proprioception

The study assessed the AEE of three groups of subjects before the intervention, after 4 weeks, 8 weeks, and 12 weeks of intervention. Each set of data followed a normal distribution, and the Mauchly sphericity test was conducted (refer to Table 7). However, the assumption of football symmetry was not satisfied ($P<0.001$), so the Greenhouse-Greisser correction method was used for degree of freedom correction. Table 8 shows that there were significant time effects, time × group interaction effects, and group effects on the degree of AEE among the three groups (all $P<0.05$). By referring to Table 9, it can be observed that the AEE values of stroke

**Table 7. Repeated measurement of Mauchly sphericity test for AEE.**

| Mauchly W value | Approximate chi-square value | Significance | Degrees of freedom | Greenhouse-Greisser correction |
|---|---|---|---|---|
| 0.454 | 67.699 | <0.001 | 5 | 0.683 |

**Table 8. Repeated measures analysis of variance for AEE.**

| Scourses | Sum of Squares | Degrees of freedom | Mean Square | F value | Significance | Eta Squared |
|---|---|---|---|---|---|---|
| Time | 1531.696 | 1.564 | 979.036 | 351.425 | <0.001 | 0.802 |
| Group×Time | 108.682 | 3.129 | 34.734 | 12.521 | <0.001 | 0.224 |
| Group | 510.554 | 2 | 255.277 | 4.372 | 0.016 | 0.091 |

**Table 9. Angle conditions of three groups of AEE at different time points.**

| Groups | Time point | Mean | Standard deviation | Confidence interval (%) | |
|---|---|---|---|---|---|
| | | | | Upper | Lower |
| CRT+BWS-TCY | before intervention | 15.138 | 4.26 | 13.52 | 16.76 |
| | 4 weeks after intervention | 11.41 | 3.84 | 9.95 | 12.87 |
| | 8 weeks after intervention | 8.714 | 3.48 | 7.39 | 10.04 |
| | 12 weeks after intervention | 7.776 | 2.99 | 6.64 | 8.91 |
| CRT | before intervention | 15.755 | 4.61 | 14.06 | 17.45 |
| | 4 weeks after intervention | 14.023 | 4.72 | 12.29 | 15.76 |
| | 8 weeks after intervention | 12.771 | 4.02 | 11.29 | 14.24 |
| | 12 weeks after intervention | 11.726 | 3.87 | 10.31 | 13.14 |
| CRT+RAT | before intervention | 15.98 | 4.62 | 14.26 | 17.70 |
| | 4 weeks after intervention | 13.523 | 3.88 | 12.07 | 14.97 |
| | 8 weeks after intervention | 11.80 | 3.77 | 10.39 | 13.21 |
| | 12 weeks after intervention | 10.283 | 3.09 | 9.13 | 11.44 |

**Table 10. Differences between AEE in three groups after intervention and before intervention (Mean±SD, °).**

| Groups | Difference between 4 weeks after intervention and before intervention | Difference between 8 weeks after intervention and before intervention | Difference between 12 weeks after intervention and before intervention |
|---|---|---|---|
| CRT +BWS-TCY | -3.73±1.77 | -6.42±3.01 | -7.36±3.05 |
| CRT | -1.73±0.87 | -2.98±1.36 | -4.03±1.65 |
| CRT+RAT | -2.46±1.21 | -4.18±1.55 | -5.69±2.16 |

**Table 11. Comparison of AEE among three groups after 12 weeks of intervention.**

| Comparison between groups | t value | P value (two-tailed) | Mean Difference | Standard error value |
|---|---|---|---|---|
| CRT+BWS-TCY *VS* CRT | -4.404 | <0.001 | -3.95 | 0.89 |
| CRT+BWS-TCY *VS* CRT+RAT | -3.164 | 0.002 | -2.51 | 0.79 |
| CRT+RAT *VS* CRT | -1.606 | 0.114 | -1.442 | 0.89 |

patients continued to decrease over time. Table 10 presents the differences between the angles after 6 weeks, 12 weeks of intervention, 8 weeks of follow-up, and before the intervention. The differences gradually decreased as time progressed. The Shapiro-Wilk test and Levine's equality of variance test were conducted to assess the normal distribution and homogeneity of variances. The results indicated that only the difference between the scores after 12 weeks of intervention and before the intervention met the requirements ($P>0.10$). Therefore, the researcher performed a two independent samples t-test to compare AEE between groups (refer to Table 11). The findings are as follows: ① The BWS-TCY group showed a significant difference compared to the CRT group ($P<0.001$), with the BWS-TCY group demonstrating better effects. ② There was a significant statistical difference between the BWS-TCY group and the RAT group (P = 0.002), with the BWS-TCY group showing better effects. ③ There was no statistical difference between the RAT group and the CRT group ($P>0.05$).

## 3.5 Results of upper limb joint mobility

The upper limb joint motion analysis (JMA) data for all three groups of subjects followed a normal distribution. The results of the Mauchly sphericity test indicated that the assumption of football symmetry was not satisfied ($P<0.001$) (see Table 12), so the Greenhouse-Greisser correction was applied. The results of the repeated measures analysis of variance are presented in Table 13. A time effect was observed in the maximum range of motion in all nine directions

**Table 12. Repeated measurement of Mauchly spherical test for the maximum range of motion of the upper limbs.**

| Outcome measures | Mauchly W value | Approximate chi-square value | Significance | Degrees of freedom | Greenhouse-Greisser correction |
|---|---|---|---|---|---|
| Shoulder flexion | 0.198 | 138.845 | <0.001 | 5 | 0.526 |
| Shoulder extension | 0.057 | 246.235 | <0.001 | 5 | 0.411 |
| Shoulder abduction | 0.157 | 158.530 | <0.001 | 5 | 0.492 |
| Shoulder adduction | 0.170 | 151.833 | <0.001 | 5 | 0.504 |
| Shoulder external rotation | 0.125 | 177.979 | <0.001 | 5 | 0.479 |
| Shoulder internal rotation | 0.143 | 166.911 | <0.001 | 5 | 0.512 |
| Elbow flexion | 0.097 | 199.743 | <0.001 | 5 | 0.461 |
| Forearm pronation | 0.253 | 76.582 | <0.001 | 5 | 0.531 |
| Forearm supination | 0.106 | 192.163 | <0.001 | 5 | 0.454 |

**Table 13. Repeated measures analysis of variance for upper limb joint mobility.**

| Outcome measures | Scourses | Sum of Squares | Degrees of freedom | Mean Square |
|---|---|---|---|---|
| Shoulder flexion | Time | 131997.529 | 303.560 | <0.001 |
| | Group | 32683.145 | 4.883 | 0.010 |
| | Group×Time | 13745.799 | 15.806 | <0.001 |
| Shoulder extension | Time | 8655.856 | 177.055 | <0.001 |
| | Group | 152.737 | 0.229 | 0.796 |
| | Group×Time | 236.172 | 2.415 | 0.082 |
| Shoulder abduction | Time | 149861.850 | 345.13 | <0.001 |
| | Group | 28084.535 | 4.256 | 0.017 |
| | Group×Time | 1163.772 | 12.855 | <0.001 |
| Shoulder adduction | Time | 11011.475 | 360.313 | <0.001 |
| | Group | 974.023 | 2.927 | 0.059 |
| | Group×Time | 478.975 | 1.683 | 0.192 |
| Shoulder external rotation | Time | 40501.941 | 470.021 | <0.001 |
| | Group | 2017.778 | 2.182 | 0.119 |
| | Group×Time | 453.458 | 2.631 | 0.055 |
| Shoulder internal rotation | Time | 38839.871 | 363.660 | <0.001 |
| | Group | 1720.211 | 3.537 | 0.033 |
| | Group×Time | 1344.688 | 6.295 | <0.001 |
| Elbow flexion | Time | 170538.522 | 404.103 | <0.001 |
| | Group | 16362.363 | 2.783 | 0.067 |
| | Group×Time | 6933.046 | 8.214 | <0.001 |
| Forearm pronation | Time | 37400.553 | 268.468 | <0.001 |
| | Group | 2364.103 | 2.263 | 0.110 |
| | Group×Time | 787.238 | 2.825 | 0.045 |
| Forearm supination | Time | 45228.957 | 383.234 | <0.001 |
| | Group | 1296.778 | 1.401 | 0.252 |
| | Group×Time | 635.845 | 2.694 | 0.055 |

($P$<0.05), with significant effects on shoulder flexion, shoulder abduction, shoulder internal rotation, elbow flexion, and forearm pronation. An interaction effect of group × time was also found ($P$<0.05). The results of the simple effect tests indicated that after the intervention, the groups only showed significant differences in shoulder flexion, shoulder abduction, and shoulder internal rotation ($P$<0.05). As shown in Table 14, the upper limb joint mobility scores of stroke patients continued to improve over time. When comparing the groups after 12 weeks of intervention, there was no statistical difference in the maximum joint range of motion in the shoulder extension direction among the three groups. However, in the remaining eight directions of motion, there were statistically significant differences between the BWS-TCY group and the CRT group, with the BWS-TCY group showing greater improvement. Furthermore, the maximum range of motion in the directions of shoulder flexion, shoulder abduction, shoulder adduction, shoulder internal rotation, and forearm supination showed statistical significance between the BWS-TCY group and the RAT group, with the BWS-TCY group demonstrating a better effect than the RAT group.

### 3.6 Results of daily living ability

The MBI data of the three groups of subjects followed a normal distribution. However, the results of the Mauchly sphericity test did not satisfy the assumption of football symmetry

**Table 14. Maximum range of motion of upper limb joints in three groups at different time points (Mean ± SD, °).**

| Outcome measures | Groups | before intervention | weeks after intervention | weeks after intervention | 12 weeks after intervention |
|---|---|---|---|---|---|
| Shoulder flexion | CRT+BWS-TCY | 37.45±30.52 | 63.86±28.69* | 81.14±28.65* | 112.97±34.73*# |
| | CRT | 33.77±25.93 | 47.19±25.36 | 55.45±26.09 | 68.48±31.83 |
| | CRT+RAT | 44.63±32.54 | 59.17±32.12 | 72.83±34.19* | 92.00±36.32* |
| Shoulder extension | CRT+BWS-TCY | 17.93±9.28 | 25.21±9.73 | 29.86±10.73 | 33.93±14.08 |
| | CRT | 19.58±8.99 | 23.03±8.48 | 27.06±7.84 | 30.87±7.70 |
| | CRT+RAT | 19.33±9.66 | 24.30±9.17 | 28.27±9.53 | 31.57±10.86 |
| Shoulder abduction | CRT+BWS-TCY | 37.14±30.75 | 65.69±28.56* | 82.69±28.59* | 113.31±33.23*# |
| | CRT | 33.42±25.93 | 50.10±26.03 | 58.55±26.69 | 72.39±32.74 |
| | CRT+RAT | 40.37±34.44 | 62.20±30.64 | 75.50±32.93* | 94.00±34.54* |
| Shoulder adduction | CRT+BWS-TCY | 14.21±7.98 | 22.00±6.40 | 27.17±5.71* | 33.69±5.45*# |
| | CRT | 13.16±7.01 | 19.29±6.25 | 22.65±5.96 | 25.90±6.20 |
| | CRT+RAT | 15.23±9.74 | 20.27±7.76 | 23.87±8.87 | 28.23±7.09* |
| Shoulder external rotation | CRT+BWS-TCY | 20.97±12.18 | 33.45±9.79 | 44.03±10.94* | 53.55±9.38* |
| | CRT | 18.00±11.51 | 29.13±10.97 | 36.94±11.76 | 45.45±13.84 |
| | CRT+RAT | 22.40±13.38 | 33.77±11.88 | 40.67±11.62 | 48.53±12.31 |
| Shoulder internal rotation | CRT+BWS-TCY | 24.52±13.15 | 37.97±9.83 | 46.21±10.26* | 60.34±9.81*# |
| | CRT | 21.32±12.29 | 33.29±10.09 | 38.74±11.32 | 45.91±13.69 |
| | CRT+RAT | 26.13±14.51 | 37.53±11.79 | 43.30±12.75 | 51.57±13.30 |
| Elbow flexion | CRT+BWS-TCY | 28.97±29.49 | 59.45±26.51* | 77.30±27.70* | 103.93±28.69* |
| | CRT | 27.26±24.73 | 47.13±24.78 | 58.48±27.02 | 72.74±33.39 |
| | CRT+RAT | 33.00±32.85 | 55.10±30.01 | 70.83±29.77 | 91.57±31.43* |
| Forearm pronation | CRT+BWS-TCY | 28.59±14.09 | 38.52±11.33 | 48.79±11.93* | 61.62±14.05* |
| | CRT | 25.68±12.96 | 34.81±9.89 | 41.90±11.48 | 50.65±13.79 |
| | CRT+RAT | 30.03±14.35 | 38.57±11.28 | 46.33±13.09 | 54.80±15.14 |
| Forearm supination | CRT+BWS-TCY | 20.62±12.85 | 33.72±10.10 | 43.79±10.39 | 55.83±13.28*# |
| | CRT | 18.42±11.46 | 30.45±8.57 | 38.68±10.41 | 48.06±13.75 |
| | CRT+RAT | 21.93±13.38 | 31.43±11.60 | 40.27±13.05 | 48.20±14.39 |

* Compared with the CRT group, the difference was statistically significant ($P< 0.05$).

# Compared with the CRT+RAT group, the difference is statistically significant. ($P< 0.05$).

($P<0.001$) (see Table 15), so the Greenhouse-Greisser correction was applied. As shown in Table 16, there was a significant time effect and a time × group interaction effect ($P<0.001$) on the MBI scores of the three groups, but no significant group effect was observed($P = 0.169$). Considering Table 17, it can be observed that the MBI scores of stroke patients continued to improve over time. The differences between the scores after 4 weeks, 8 weeks, and 12 weeks of intervention and before the intervention are presented in Table 18. These differences gradually increased with the passage of time. A two independent samples t-test was conducted to compare the groups after 12 weeks of intervention (see Table 19): ① Comparing the BWS-TCY group and the CRT group, a significant difference was found between the two groups

**Table 15. Repeated measurement of Mauchly sphericity hypothesis test for MBI.**

| Mauchly W value | Approximate chi-square value | Significance | Degrees of freedom | Greenhouse-Greisser correction |
|---|---|---|---|---|
| 0.540 | 52.786 | <0.001 | 5 | 0.744 |

**Table 16. Repeated measures analysis of variance of MBI in three groups.**

| Scourses | Sum of Squares | Degrees of freedom | Mean Square | F value | Significance | Eta Squared |
|---|---|---|---|---|---|---|
| Time | 28081.49 | 2.232 | 12578.59 | 533.25 | <0.001 | 0.86 |
| Group×Time | 1416.98 | 4.465 | 317.35 | 13.45 | <0.001 | 0.24 |
| Group | 1682.47 | 2 | 841.23 | 1.812 | 0.169 | 0.04 |

**Table 17. MBI scores of three groups at different time points.**

| Groups | Time point | Mean | Standard deviation | Confidence Interval (%) | |
|---|---|---|---|---|---|
| | | | | Upper | Lower |
| CRT+BWS-TCY | before intervention | 46.38 | 14.01 | 41.05 | 51.71 |
| | 4 weeks after intervention | 55.86 | 12.61 | 51.06 | 60.66 |
| | 8 weeks after intervention | 66.21 | 13.54 | 61.06 | 71.36 |
| | 12 weeks after intervention | 76.55 | 12.54 | 71.78 | 81.32 |
| CRT | before intervention | 47.74 | 9.99 | 44.08 | 51.01 |
| | 4 weeks after intervention | 52.42 | 9.56 | 48.91 | 55.93 |
| | 8 weeks after intervention | 59.81 | 9.41 | 56.36 | 63.26 |
| | 12 weeks after intervention | 64.52 | 10.83 | 60.54 | 68.49 |
| CRT+RAT | before intervention | 48.33 | 10.61 | 44.37 | 52.30 |
| | 4 weeks after intervention | 55.33 | 10.98 | 51.23 | 59.43 |
| | 8 weeks after intervention | 63.67 | 11.29 | 59.45 | 67.88 |
| | 12 weeks after intervention | 71.67 | 10.37 | 67.80 | 75.54 |

**Table 18. Difference between three groups after MBI intervention and before intervention (Mean±SD, score).**

| Groups | Difference between 4 weeks after intervention and before intervention | Difference between 8 weeks after intervention and before intervention | Difference between 12 weeks after intervention and before intervention |
|---|---|---|---|
| CRT +BWS-TCY | 8.59±7.41 | 9.48±3.86 | 30.17±6.78 |
| CRT | 6.29±7.99 | 4.68±4.82 | 16.77±6.78 |
| CRT+RAT | 7.03±5.35 | 7.00±5.81 | 23.33±7.69 |

**Table 19. Comparison of MBI between three groups after 12 weeks of intervention.**

| Comparison between groups | t value | P value (two-tailed) | Mean Difference | Standard error value |
|---|---|---|---|---|
| CRT+BWS-TCY *VS* CRT | 6.98 | <0.001 | 13.39 | 1.92 |
| CRT+BWS-TCY *VS* CRT+RAT | 3.32 | 0.002 | 6.84 | 2.05 |
| CRT+RAT *VS* CRT | 3.54 | 0.001 | 6.56 | 1.86 |

($P<0.001$), with the BWS-TCY group showing better results than the CRT group; ② Comparing the BWS-TCY group with the RAT group, a significant statistical difference was observed between the two groups (P = 0.002), with the BWS-TCY group demonstrating better effects than the RAT group; ③ Comparing the RAT group with the CRT group, a significant statistical difference was found between the two groups (P = 0.001), with the RAT group showing better effects than the CRT group.

**Table 20. Repeated measurement of Mauchly sphericity hypothesis test for SS-QOL.**

| Mauchly W value | Approximate chi-square value | Significance | Degrees of freedom | Greenhouse-Greisser correction |
|---|---|---|---|---|
| 0.712 | 29.177 | <0.001 | 5 | 0.814 |

**Table 21. Repeated measures analysis of variance of SS-QOL among three groups.**

| Scourses | Sum of Squares | Degrees of freedom | Mean Square | F value | Significance | Eta Squared |
|---|---|---|---|---|---|---|
| Time | 25933.841 | 2.443 | 10617.002 | 392.434 | <0.001 | 0.819 |
| Group×Time | 854.587 | 4.885 | 174.929 | 6.466 | <0.001 | 0.206 |
| Group | 1914.391 | 2 | 957.196 | 2.707 | 0.072 | 0.059 |

**Table 22. SS-QOL scores of three groups at different time points.**

| Groups | Time point | Mean | Standard deviation | Confidence Interval (%) | |
|---|---|---|---|---|---|
| | | | | Upper | Lower |
| CRT+BWS-TCY | before intervention | 99.79 | 13.81 | 94.54 | 105.05 |
| | 4 weeks after intervention | 110.34 | 11.36 | 106.02 | 114.67 |
| | 8 weeks after intervention | 114.59 | 11.82 | 110.09 | 119.08 |
| | 12 weeks after intervention | 127.55 | 9.46 | 123.95 | 131.15 |
| CRT | before intervention | 99.58 | 10.65 | 96.56 | 106.44 |
| | 4 weeks after intervention | 104.61 | 8.75 | 101.40 | 107.82 |
| | 8 weeks after intervention | 108.84 | 8.27 | 105.81 | 111.87 |
| | 12 weeks after intervention | 117.32 | 6.72 | 114.86 | 119.79 |
| CRT+RAT | before intervention | 99.37 | 13.02 | 94.51 | 104.23 |
| | 4 weeks after intervention | 109.37 | 9.45 | 105.83 | 112.89 |
| | 8 weeks after intervention | 111.23 | 9.89 | 109.75 | 113.89 |
| | 12 weeks after intervention | 124.77 | 9.38 | 121.14 | 125.06 |

## 3.7 Quality of life

Through four repeated measurements of SS-QOL, Mauchly's spherical symmetry assumption was not met ($P<0.05$) (refer to Table 20). Therefore, the Greenhouse-Greisser correction results were utilized. The results of the analysis of variance indicated (see Table 21) that both the time effect and the interaction effect of time × group effect were statistically significant ($P<0.001$). However, there was no statistical difference observed in the group effect (P = 0.072). As shown in Table 22, the SS-QOL scores of stroke patients continue to improve as time progresses. The inter-group comparison revealed no statistical significance among the three patient groups across the four time periods.

## 4. Discussion

### 4.1 Effect of BWS-TCY training on upper limb motor function in stroke patients

Upper limb motor dysfunction after stroke is closely associated with a decrease in excitability of the injured cerebral cortex and abnormal innervation of the limbs [48, 49]. Repeated movement training of the affected limb can expand the corresponding cortical representative area and enhance nerve signal transmission efficiency [50]. This directly affects the movement and

sensation of the upper limbs after a stroke. When the brain loses motor control, it can result in difficulty or inability to move voluntarily. In this study, two indicators, FMA-UE and WMFT, were used to evaluate the motor function of the patients. The results demonstrated a time effect in all three patient groups, indicating improvement in upper limb motor function over time. Furthermore, there was an interaction and group effect of time and group. Further analysis revealed that the effect of BWS-TCY was superior to simple CRT, although there was no statistical significance when compared to RAT, suggesting that both intervention methods showed promising recovery effects. It is speculated that the similarity in effects between BWS-TCY and RAT may be attributed to the advantages of upper limb rehabilitation robots. Upper limb robot training can enhance the fluency and accuracy of patients' upper limb movements. Additionally, force feedback technology can be utilized to adjust power or resistance in real time, guiding the patient's upper limb movements. Audio-visual feedback can also be used to attract the patient's attention and increase participation [51]. The upper limb rehabilitation robot utilizes a forearm support to enable patients to undergo training without the interference of gravity on active upper limb movements [52]. This allows patients to fully engage in a wide range of upper limb movement training at an early stage, promoting the balance between agonist and antagonist muscles. The robot also coordinates muscle tension and prevents abnormal movements, facilitating isolated movements for the patients. Moreover, it overcomes the limitations of relying solely on the therapist's experience and expertise, enhances the effectiveness of the training, and significantly improves training efficiency by eliminating the monotony associated with conventional rehabilitation training.

At the same time, TCY also exerts its significant advantages. It involves coordinated exercises of both upper limbs, which can facilitate the remodeling of the cortex and nerves, leading to an improvement in the motor function of stroke patients' upper limbs. Furthermore, the simultaneous movement of both upper limbs can activate sensory and motor areas on both sides of the body, resulting in an enhancement of sensory performance in the affected limbs. TCY is considered an isotonic exercise that targets all joints and ligaments, thereby increasing muscle strength and improving the biomechanical structure of joints. Moreover, it positively impacts the function of the central motor system, muscle spindles, and tendon organs after a stroke [53]. Additionally, TCY helps regulate abnormal muscle strength and tension, ultimately improving muscle and joint movement.

## 4.2 Effect of BWS-TCY training on proprioception in stroke patients

Stroke has a significant impact on the movement and sensation of the upper limbs [50]. When the brain loses motor control, it can lead to difficulties in movement or even prevent voluntary movement. This impairs the dexterity and coordination of the upper limbs, causes sensory and proprioceptive disorders, and reduces the ability to control limb position and movement. Proprioceptive deficit is a common sensory disorder following stroke, with approximately 50% of patients experiencing loss of upper limb proprioception [54]. In this study, a robot was used to measure the absolute angle error of the patient's elbow joint, providing an easy and fast method of operation. The results revealed a time effect within the three patient groups, as well as an interaction effect between time and group. When comparing the three groups, it was found that the rehabilitation effect of BWS-TCY was superior to that of RAT and CRT. This suggests that the unique TCY movement may be the primary reason for enhancing proprioception. During TCY exercise, the upper limbs undergo alternating left and right rotations, incorporating virtual and actual changes to promote coordination between the upper and lower limbs and effective coordination of joint muscles. This can ultimately improve proprioception and coordination in post-stroke patients [36]. Coordination disorder is a prominent

issue in stroke patients, and the restoration of proprioception is beneficial for postural control and further influences the recovery of upper limb motor function [55, 56]. TCY emphasizes the cultivation of the spirit throughout the body in order to achieve a state of systemic relaxation. This relaxation can help relieve muscle tension and improve the compensatory and reorganization ability of the central nervous system. Additionally, TCY encourages individuals to actively engage and strengthen their subjective initiative, which aids in the input and enhancement of proprioception [36, 57]. During TCY movements, the upper limbs perform asymmetrical movements such as forearm pronation and supination, shoulder joint abduction, and wrist joint dorsiflexion. These movements resemble the symmetrical and bilateral asymmetrical patterns observed in the neuromuscular proprioceptive facilitation technique, which helps in the restoration of proprioception [58]. The upper limb rehabilitation robot is designed to provide stroke patients with a specific training situation to relearn movements. The training process emphasizes the importance of feedback for movement control. By offering visual and auditory feedback, real-time training information can be obtained to correct abnormal movement patterns, learn correct movement control [59], and increase the proprioceptive input of movement through highly repetitive and engaging upper limb movements. This helps promote the brain to generate movement plans, effectively improving the patient's upper limb movement ability, and mobilizing their enthusiasm for movement [32]. The upper limb robot utilizes advanced motion tracking technology to display the real-time trajectory curve of the patient's upper limb movement [31]. This allows the patient to perceive the spatial position and motion status of their upper limb using the trajectory curve, enhancing proprioceptive input and facilitating conscious movement adjustments [52, 60].

### 4.3 Effect of BWS-TCY training on joint mobility in stroke patients

This study aimed to evaluate the joint mobility of the upper limbs in stroke patients. The results demonstrated that all three patient groups showed an improvement in joint mobility over time. Specifically, there were statistically significant differences in shoulder flexion, shoulder abduction, shoulder internal rotation, elbow flexion, and forearm pronation when comparing the groups over time. Additionally, the BWS-TCY group exhibited better improvement in shoulder flexion, shoulder abduction, shoulder adduction, shoulder internal rotation, and forearm supination compared to the CRT group. The effectiveness of the BWS-TCY intervention in improving joint mobility is attributed to the circular movements involved in TCY exercises. These exercises enhance flexibility and range of motion in the affected shoulder joint. Furthermore, the incorporation of chest and back exercises during TCY promotes the recovery of muscle strength and function in the pectoralis major and deltoid muscles [61].

### 4.4 Effect of BWS-TCY training on daily self-care ability and quality of life of stroke patients

The main objective of rehabilitation treatment is to help patients achieve independent living, self-care, and reintegrate into their families and society as soon as possible. In this study, the researchers used the MBI scale to assess the patients' ability to perform daily activities. The results indicated that there was a significant time effect within each of the three groups of stroke patients. Furthermore, there was an interaction effect of time × group among the three groups. By comparing the groups, it was found that there was no significant difference between the BWS-TCY group and the RAT group, indicating that both intervention methods had similar effects on daily living activities. However, both groups showed significant differences compared to the CRT group. The researchers also assessed the quality of life using the SS-QOL scale. The results showed a significant time effect in all three groups, indicating that the quality

of life continued to improve over time. There was also an interaction effect of time × group, but no significant group effect was observed, suggesting that there was no statistical difference between the two groups. According to Huang et al., BWS-TC has been found to effectively enhance daily living abilities and is considered superior to CRT [40]. The differences between the two groups may be attributed to the distinctive movements of TCY and the high-repetition and high-intensity exercises performed with the rehabilitation robot. The CRT process not only involves physical therapy but also incorporates functional activities of daily living, such as face washing, drinking from a cup, eating, and hair combing. Patients are consistently encouraged and supervised to actively participate and cooperate. TCY training involves an arc movement from the inside out and bottom up, which aligns with certain daily life actions like drinking from a cup and washing the face. This independent training method is suitable for patients and offers stroke patients opportunities for ongoing functional exercise. Enhancing self-care abilities in daily life will undoubtedly contribute to the improvement of overall quality of life.

## 4.5 Security analysis

Before implementing the intervention, patients and their families were informed about the intervention plan, including the measures, intensity, frequency, and duration. Their approval and cooperation were obtained, and they were also informed about the possible discomfort they may experience. The subjects were closely monitored throughout the 12-week intervention period, and no adverse reactions were observed. It is recommended to carefully select the indications for the trial and closely observe any discomfort reactions during the intervention process.

## 4.6 Limitations and prospects

This study also has some limitations. Firstly, only the outcome assessors and statistical analysts were blinded to reduce bias, while it was not possible to blind the subjects and therapists. Secondly, the subjects were not required to record their daily exercise and treatment status, such as exercise type and medication use, during the intervention period. These factors may have influenced the final results. Thirdly, there was no real-time monitoring during the follow-up period, which could have been affected by patients' spontaneous movements. Fourthly, the intensity of the three intervention approaches was not examined, which could have influenced the results. Fifthly, the sample size of this study is small and it is a single-center study. To verify the results, future studies should expand the sample size to multiple centers. Sixthly, it should be noted that TCY training is not only about movements. Many studies on Tai Chi have mentioned the role of motor imagery, but this aspect was not explored in this study. Seventhly, the patient follow-up data were not evaluated and analyzed, and an 8-week follow-up could be included in future research. Finally, the mechanism of action of BWS-TCY has not been explored from the perspective of biomechanics and brain science. In future studies, near-infrared brain function testing and functional magnetic resonance technology can be utilized to investigate the mechanism of action of BWS-TCY on stroke patients.

## 5. Conclusion

The pilot study of 15 patients was generally consistent with our assumptions in the actual study. The 12-week BWS-TCY intervention has been shown to effectively enhance upper limb motor function, proprioception, joint mobility, daily living ability, and quality of life. The BWS-TCY intervention demonstrated superior effects on upper limb motor function,

proprioception, JMA (excluding shoulder extension), and daily living ability when compared to CRT. Additionally, when compared to RAT, the BWS-TCY intervention showed greater improvements in proprioception, JMA (including shoulder forward flexion, shoulder abduction, shoulder joint internal and external rotation, and forearm supination direction), and daily living ability.

## Supporting information

**S1 Checklist. CONSORT 2010 checklist of information to include when reporting a randomised trial\*.**
(DOC)

**S1 Raw data. Raw data required to replicate the results.**
(DOCX)

**S1 File. Trial protocol.**
(DOCX)

**S2 File.**
(DOCX)

## Acknowledgments

We sincerely thank the authors of the primary studies who provided the data and each of the researchers involved in this study.

## Author Contributions

**Conceptualization:** Jiening Wang, Naizhen Wang.

**Data curation:** Liying Zhang, Xiaoming Yu, Wangsheng Liao, Jiening Wang.

**Formal analysis:** Liying Zhang, Xiaoming Yu.

**Funding acquisition:** Xiaoming Yu, Jiening Wang, Zhou Huanxia.

**Investigation:** Xiaoming Yu, Zhou Huanxia.

**Methodology:** Xiaoming Yu, Jiening Wang, Naizhen Wang, Zhou Huanxia.

**Project administration:** Xiaoming Yu, Wangsheng Liao, Yan Lu, Naizhen Wang, Zhou Huanxia.

**Resources:** Liying Zhang, Wangsheng Liao, Yan Lu, Zhou Huanxia.

**Software:** Liying Zhang.

**Supervision:** Liying Zhang, Wangsheng Liao, Jiening Wang, Naizhen Wang, Zhou Huanxia.

**Visualization:** Jiening Wang, Zhou Huanxia.

**Writing – original draft:** Liying Zhang, Xiaoming Yu.

**Writing – review & editing:** Jiening Wang, Naizhen Wang, Zhou Huanxia.

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
