## [Decision Letter · Decision Letter 0]

5 Jun 2024

PONE-D-24-10912Effects of Body Weight-supported Tai Chi Yunshou training on upper limb motor function in stroke patients: A three-arm parallel randomized controlled trialPLOS ONE

Dear Dr. Wang,

Thank you for submitting your manuscript to PLOS ONE. After careful consideration, we feel that it has merit but does not fully meet PLOS ONE’s publication criteria as it currently stands. Therefore, we invite you to submit a revised version of the manuscript that addresses the points raised during the review process.

We look forward to receiving your revised manuscript.

Kind regards,

Domiziano Tarantino, MD

Academic Editor

PLOS ONE

Journal Requirements:

This study was founded by the Shanghai Municipal Health and Health Commission Chinese Medicine Research Project (Grant No. 2020LP004), Fujian Provincial Clinical Medical Research Center for First Aid and Rehabilitation in Orthopaedic Trauma(Grant No. 2020Y2014), Medical Discipline Construction Project of Pudong New Area Commission of Health and Family Planning (Grant No. PW2022A-71), Pudong New Area “National Traditional Chinese Medicine Development and Reform Pilot” Construction Project (Grant No. PDZY-2022-0702) and Excellent Young Medical Talents Training Program of Pudong New District Health Commission (Grant No. PWRq2020-13).

Additional Editor Comments:

Dear Authors,

please revise the manuscript according to the suggestions provided by the Reviewers.

Reviewers' comments:

Reviewer's Responses to Questions

**Comments to the Author**

1. Is the manuscript technically sound, and do the data support the conclusions?

Reviewer #1: No

Reviewer #2: Yes

Reviewer #3: Partly

2. Has the statistical analysis been performed appropriately and rigorously? 

Reviewer #1: Yes

Reviewer #2: Yes

Reviewer #3: Yes

3. Have the authors made all data underlying the findings in their manuscript fully available?

Reviewer #1: Yes

Reviewer #2: Yes

Reviewer #3: Yes

4. Is the manuscript presented in an intelligible fashion and written in standard English?

Reviewer #1: Yes

Reviewer #2: Yes

Reviewer #3: Yes

5. Review Comments to the Author

**Reviewer #1:** This randomized control study evaluated the efficacy of BWS-TCY, RAT, CRT among subacute (within 6 months after onset) stroke patients. The rater was blinded to give an unbiased rating. The physicians and patients were not blinded, but this was reasonable when the nature of the protocols were considered. As authors mentioned in the introduction, enhance recovery after stroke is of importance. There is no question in the rationale of this study, and the impact of the results is more than huge. However, the reviewer has to pose a question regarding the robustness of the study and cannot accept all the data because this manuscript contains a lot of inconsistencies represented as follows:

1. Abstract Results. "The AAE of the BWS-TCY group showed no statistical difference (P<0.05) when compared to the CRT group and RAT group (P>0.05)." Is P<0.05 not significant?

2. Abstract Results. "However, there was a statistical difference (P>0.05) in MBI between the two groups (P<0.01)." Is P>0.05 significant?

3. 3.2.1 Comparison of the basic conditions of the three groups before intervention. "indicating that the baselines were comparable in terms of sex (refer to Table 1)." Table 1 is not showing the baselines across sexes.

4. 2.4 Sample size. "Sample size calculation was conducted for the main objective, which focused on the treatment effect 4 weeks after surgery." The reviewer cannot figure out why the sample size was estimated using patients after surgery although this study targeted patients with stroke.

5. 3.6 Results of Daily living ability. "As shown in Table 17, there was a significant time effect and a time × group interaction effect (P<0.001) on the MBI scores of the three groups," In table 17, GroupxTime has Significance of 0.169?

6. 3.7 Quality of Life. "The results of the analysis of variance indicated (see Table 22) that both the time effect and the interaction effect of time × group were statistically significant (P<0.001)."In table 22, GroupxTime has Significance of 0.072?

**Reviewer #2:** This is a well-written report of a clinical trial of tai chi on limb function in stroke patients. The study is well-designed and obviously is cannot be double masked, but some thought has been put into ascertainment bias, and the study's limitations are outlined. Some statistical comments:

1. The randomization procedure is not described. Stating that the data are "randomly split by a statistician" is meaningless. Permuted blocks? Random blocks? Complete randomization?

2. Sample size is based on a single primary outcome in 4 weeks, and yet it seems you are much more interested in longitudinal assessment, and multiple, possibly correlated outcomes. Your "significance is set at 0.05" of course has no meaning when you are testing so many hypotheses. You either should be doing a Holm's adjustment or giving a long laundry list of exploratory analyses for which the study is not powered. You can't have it both ways.

3. CONSORT no longer recommends p-values for baseline characteristics for two reasons: multiple testing means that 1/20 will be out of balance just by chance; failure to reject the null does not prove they are similar.

4. The conclusions should state whether the extensive sample size assumptions you made based on a pilot of 15 patients were actually realized in the trial.

**Reviewer #3: **I congratulate the authors for the rigorous and well applied statistical analysis and the study design. However, I advise some revisions and suggestions in order to publish this article

Introduction

These sentences are repeated in very similar way and are very near; it could be possible that at least this paragraph was AI generated

128-134 Previous studies have demonstrated that body weight supported Tai Chi gait training, using a suspension device within a balance bar, can enhance lower limb motor function and balance in early-stage stroke patients. However, this type of training typically requires the assistance of two therapists simultaneously, which can be time-consuming and labor-intensive. Rehabilitation robots, equipped with exoskeletons and robotic arms, have the capability to provide auxiliary, resistance, and passive training

142-149 By incorporating Tai Chi with suspension devices, patients who are unable to fully bear weight can initiate training promptly. However, this training method requires the presence of at least two therapists simultaneously, resulting in time and labor consumption. Rehabilitation robots, on the other hand, offer gravity compensation and are user-friendly[40]. Rehabilitation robotic exoskeletons and joysticks not only provide gravity compensation but also offer power-assisted training, esistance training, and passive training.

Study method

Some references are clearly wrong: ie

"The diagnostic standards for stroke in Western medicine refer to the

192 'Cerebrovascular Disease Diagnostic Criteria' of the 1995 National Cerebrovascular

193 Disease Academic Conference[43], which require confirmation through CT or MRI

194 examination. On the other hand, the diagnostic standards for stroke in Traditional

195 Chinese Medicine are based on the 'Standards for Diagnosis and Treatment of

196 Diseases' formulated by the traditional Chinese medicine industry of the People's

197 Republic of China (1995 edition) [44]."

"[43] THOMAS L, COUPE J, CROSS L, et al. Interventions for treating urinary incontinence

910 after stroke in adults [J]. The Cochrane database of systematic reviews, 2019, 2(2):

911 CD004462.

912 [44] YEH G, CHAN C, WAYNE P, et al. The Impact of Tai Chi Exercise on Self-Efficacy,

913 Social Support, and Empowerment in Heart Failure: Insights from a Qualitative Sub-Study

914 from a Randomized Controlled Trial [J]. PloS one, 2016, 11(5): e0154678."

Refering to sample size the authors wrote "o. Sample size calculation

245 was conducted for the main objective, which focused on the treatment effect 4 weeks

246 after surgery." It is not clear which surgery they refer to

Results

I suggest to include baseline NIHSS in order to evaluate the severity of stroke among the partecipant and their comparability

Conclusion

BWS-TCY is clearly better than CRT only but I think that the affirmation “Furthermore, the improvement effect surpasses that of RAT is not supported by the data” (i.e. there are not any difference in SSQOL between BWS-TCY and RAT and the mBI after 12 weeks is more than 60 in all the three groups)

Supporting information

In trial protocol parts of the text is clearly copied from the article (or viceversa) with the same mistakes in the references

6. PLOS authors have the option to publish the peer review history of their article (what does this mean?). If published, this will include your full peer review and any attached files.

Reviewer #1: No

Reviewer #2: No

Reviewer #3: **Yes: **Andrea Plutino

---

## [Author Response · Author response to Decision Letter 0]

8 Jul 2024

Dear editor and dear reviewers:

Thank you for your letter and for the reviewers’ comments concerning our manuscript entitle “Effects of Body Weight-supported Tai Chi Yunshou training on upper limb motor function in stroke patients: A three-arm parallel randomized controlled trial” (ID: PONE-D-24-10912). Those comments are all valuable and very helpful for revising and improving our paper, as well as the important guiding significance to our researches. We have studied comments carefully and have made correction which we hope meet with approval. Revised portion are marked in red in the paper. The main corrections in the paper and the respondse to the reviewer’s comments are as following:

Respondse to reviewers:

Reviewer #1:

1. Comment: Abstract Results. "The AAE of the BWS-TCY group showed no statistical difference (P<0.05) when compared to the CRT group and RAT group (P>0.05)." Is P<0.05 not significant?

Response: We apologize for the incorrect wording in our previous communication. Our intended message is that the average absolute error (AAE) of the BWS-TCY group did not show a statistically significant difference (P>0.05) when compared to the CRT and RAT groups.

2. Comment: Abstract Results. "However, there was a statistical difference (P>0.05) in MBI between the two groups (P<0.01)." Is P>0.05 significant?

Response: We appreciate your reminder and apologize for any confusion caused by our previous communication. Our intended message is that there was a statistically significant difference (P<0.01) in MBI between the two groups.

3. Comment: 3.2.1 Comparison of the basic conditions of the three groups before intervention. "indicating that the baselines were comparable in terms of sex (refer to Table 1)." Table 1 is not showing the baselines across sexes.

Response: Thanks for your kind reminder. We are very sorry that the wording was wrong due to our carelessness. What we need to express is that the baseline is comparable.

4. Comment: 2.4 Sample size. "Sample size calculation was conducted for the main objective, which focused on the treatment effect 4 weeks after surgery." The reviewer cannot figure out why the sample size was estimated using patients after surgery although this study targeted patients with stroke. 

Response: Thanks for your kind reminder. We are very sorry that the wording was wrong due to our carelessness. We need to express that it should be four weeks after the rehabilitation intervention.

5. Comment: 3.6 Results of Daily living ability. "As shown in Table 17, there was a significant time effect and a time × group interaction effect (P<0.001) on the MBI scores of the three groups," In table 17, GroupxTime has Significance of 0.169?

Response: Thank you for your kind reminder and sorry for our carelessness. By querying the original data, we found that we confused the data on group effects and interaction effects when tabulating. In table 17, Group effect has Significance of 0.169.

6. Comment: 3.7 Quality of Life. "The results of the analysis of variance indicated (see Table 22) that both the time effect and the interaction effect of time × group were statistically significant (P<0.001)."In table 22, GroupxTime has Significance of 0.072?

Response: Thank you for your kind reminder and sorry for our carelessness. By querying the original data, we found that we confused the data on group effects and interaction effects when tabulating. In table 22, Group effect has Significance of 0.072.

Reviewer #2: 

1. Comment: The randomization procedure is not described. Stating that the data are "randomly split by a statistician" is meaningless. Permuted blocks? Random blocks? Complete randomization? 

Response: Thanks for your kind reminder. Patients were assigned treatment using a complete randomization scheme where the randomization procedure was conducted through a software that utilized random permuted blocks. The subjects, totaling 93, were evenly distributed into three groups: BWS-TCY group, CRT group, and RAT group, each consisting of 31 cases. The random sequence was created by an independent professional statistician using SPSS software (IBM Corp., IBM SPSS Statistics, V25, Armonk, NY, USA), with the random number seed set to 20210608. 

2. Comment: Sample size is based on a single primary outcome in 4 weeks, and yet it seems you are much more interested in longitudinal assessment, and multiple, possibly correlated outcomes. Your "significance is set at 0.05" of course has no meaning when you are testing so many hypotheses. You either should be doing a Holm's adjustment or giving a long laundry list of exploratory analyses for which the study is not powered. You can't have it both ways. 

Response: The sample size calculation has fully taken into account the repeatability of longitudinal data, and has been calculated using G*Power software. The calculation process can be referred to the figure below. We have explained in the statistical analysis section that 0.05 is used as the test level and the Holm's adjustment P value is used as the basis for significance. In the result description section, it is supplementary to explain whether the results after Holm's adjustment are significant.

3. Comment: CONSORT no longer recommends p-values for baseline characteristics for two reasons: multiple testing means that 1/20 will be out of balance just by chance; failure to reject the null does not prove they are similar. 

Response: Thank you for your kind reminder. We no longer use P values to compare baseline characteristics of patients in randomized controlled trials. Instead, we directly display the baseline characteristics of the three groups and use clinical experience to determine whether there are clinically significant differences.

4. Comment: The conclusions should state whether the extensive sample size assumptions you made based on a pilot of 15 patients were actually realized in the trial. 

Response: Thanks for the reminder, we have stated in the Conclusion that the broad sample size assumptions we made based on a pilot of 15 patients were actually realized in the trial.

Reviewer #3: 

1. Comment: Introduction

These sentences are repeated in very similar way and are very near; it could be possible that at least this paragraph was AI generated

128-134 Previous studies have demonstrated that body weight supported Tai Chi gait training, using a suspension device within a balance bar, can enhance lower limb motor function and balance in early-stage stroke patients. However, this type of training typically requires the assistance of two therapists simultaneously, which can be time-consuming and labor-intensive. Rehabilitation robots, equipped with exoskeletons and robotic arms, have the capability to provide auxiliary, resistance, and passive training

142-149 By incorporating Tai Chi with suspension devices, patients who are unable to fully bear weight can initiate training promptly. However, this training method requires the presence of at least two therapists simultaneously, resulting in time and labor consumption. Rehabilitation robots, on the other hand, offer gravity compensation and are user-friendly[40]. Rehabilitation robotic exoskeletons and joysticks not only provide gravity compensation but also offer power-assisted training, esistance training, and passive training.

Response: Thank you for your kind reminder and sorry for our carelessness, we have removed the duplicate content (Line 128-134).

2. Comment: Study method

Some references are clearly wrong: ie

"The diagnostic standards for stroke in Western medicine refer to the

192 'Cerebrovascular Disease Diagnostic Criteria' of the 1995 National Cerebrovascular

193 Disease Academic Conference[43], which require confirmation through CT or MRI

194 examination. On the other hand, the diagnostic standards for stroke in Traditional

195 Chinese Medicine are based on the 'Standards for Diagnosis and Treatment of

196 Diseases' formulated by the traditional Chinese medicine industry of the People's

197 Republic of China (1995 edition) [44]."

"[43] THOMAS L, COUPE J, CROSS L, et al. Interventions for treating urinary incontinence

910 after stroke in adults [J]. The Cochrane database of systematic reviews, 2019, 2(2):

911 CD004462.

912 [44] YEH G, CHAN C, WAYNE P, et al. The Impact of Tai Chi Exercise on Self-Efficacy,

913 Social Support, and Empowerment in Heart Failure: Insights from a Qualitative Sub-Study

914 from a Randomized Controlled Trial [J]. PloS one, 2016, 11(5): e0154678."

Response: Thank you for your kind reminder. Our diagnostic criteria for stroke should be found through CT/MRI examination, including hemorrhagic stroke and ischemic stroke. Therefore no references were involved.

3. Comment: Refering to sample size the authors wrote "o. Sample size calculation

245 was conducted for the main objective, which focused on the treatment effect 4 weeks

246 after surgery." It is not clear which surgery they refer to

Response: Thanks for your kind reminder. We are very sorry that the wording was wrong due to our carelessness. We need to express that it should be four weeks after the rehabilitation intervention.

4. Comment: Results

I suggest to include baseline NIHSS in order to evaluate the severity of stroke among the partecipant and their comparability 

Response: Thank you for your reminder. We retrieved their basic information from the patient database and included the NIHSS in the baseline assessment to evaluate the severity of the participants' strokes. The NIHSS data revealed no significant differences, suggesting similar baseline characteristics.

5. Comment: Conclusion

BWS-TCY is clearly better than CRT only but I think that the affirmation “Furthermore, the improvement effect surpasses that of RAT is not supported by the data” (i.e. there are not any difference in SSQOL between BWS-TCY and RAT and the mBI after 12 weeks is more than 60 in all the three groups) 

Response: Thanks for your kind reminder. Based on the statistical results, we reorganized the statements. The BWS-TCY intervention demonstrated superior effects on upper limb motor function, proprioception, JMA (excluding shoulder extension), and daily living ability when compared to CRT. Additionally, when compared to RAT, the BWS-TCY intervention showed greater improvements in proprioception, JMA (including shoulder forward flexion, shoulder abduction, shoulder joint internal and external rotation, and forearm supination direction), and daily living ability.

6. Comment: Supporting information

In trial protocol parts of the text is clearly copied from the article (or viceversa) with the same mistakes in the references 

Response: Thank you for your kind reminder. We have replaced or deleted the erroneous references.

---

## [Decision Letter · Decision Letter 1]

7 Aug 2024

PONE-D-24-10912R1Effects of Body Weight-supported Tai Chi Yunshou training on upper limb motor function in stroke patients: A three-arm parallel randomized controlled trialPLOS ONE

Dear Dr. Wang,

Thank you for submitting your manuscript to PLOS ONE. After careful consideration, we feel that it has merit but does not fully meet PLOS ONE’s publication criteria as it currently stands. Therefore, we invite you to submit a revised version of the manuscript that addresses the points raised during the review process.

We look forward to receiving your revised manuscript.

Kind regards,

Domiziano Tarantino, MD

Academic Editor

PLOS ONE

Journal Requirements:

Reviewers' comments:

Reviewer's Responses to Questions

**Comments to the Author**

1. If the authors have adequately addressed your comments raised in a previous round of review and you feel that this manuscript is now acceptable for publication, you may indicate that here to bypass the “Comments to the Author” section, enter your conflict of interest statement in the “Confidential to Editor” section, and submit your "Accept" recommendation.

Reviewer #2: (No Response)

Reviewer #3: (No Response)

2. Is the manuscript technically sound, and do the data support the conclusions?

Reviewer #2: (No Response)

Reviewer #3: Yes

3. Has the statistical analysis been performed appropriately and rigorously? 

Reviewer #2: (No Response)

Reviewer #3: Yes

4. Have the authors made all data underlying the findings in their manuscript fully available?

Reviewer #2: (No Response)

Reviewer #3: Yes

5. Is the manuscript presented in an intelligible fashion and written in standard English?

Reviewer #2: (No Response)

Reviewer #3: Yes

6. Review Comments to the Author

Reviewer #2: There are still some minor problems. The statistician who did the randomization needs to read a book on randomization, perhaps "Randomization in Clinical Trials, Theory and Practice, 2016", to understand the difference between complete randomization and random blocks. Change "complete randomization scheme" to "randomization procedure", because complete randomization is not the same thing as random blocks. For random blocks, you must state the random blocks sizes that were selected. Was the selection of random sizes equally likely?

In Section 5, just copying my suggestion into the text does not answer the question. WHAT specific assumptions from your pilot of 15 were actually made (e.g., variability, coefficient of variation), and what values were realized in the study? These are simple questions that any biostatistics students would be able to answer.

Reviewer #3: Despite the positive response by the authors my first comment was not addressed:

"Introduction

These sentences are repeated in very similar way and are very near; it could be

possible that at least this paragraph was AI generated

128-134 Previous studies have demonstrated that body weight supported Tai Chi gait

training, using a suspension device within a balance bar, can enhance lower limb

motor function and balance in early-stage stroke patients. However, this type of

training typically requires the assistance of two therapists simultaneously, which can

be time-consuming and labor-intensive. Rehabilitation robots, equipped with

exoskeletons and robotic arms, have the capability to provide auxiliary, resistance,

and passive training

142-149 By incorporating Tai Chi with suspension devices, patients who are unable to

fully bear weight can initiate training promptly. However, this training method

requires the presence of at least two therapists simultaneously, resulting in time and

labor consumption. Rehabilitation robots, on the other hand, offer gravity

compensation and are user-friendly[40]. Rehabilitation robotic exoskeletons and

joysticks not only provide gravity compensation but also offer power-assisted training,

esistance training, and passive training.

Response: Thank you for your kind reminder and sorry for our carelessness, we have

removed the duplicate content (Line 128-134)."

Furthermore I kindly ask why the founding and the grant numbers were changed from the previous text

7. PLOS authors have the option to publish the peer review history of their article (what does this mean?). If published, this will include your full peer review and any attached files.

Reviewer #2: No

Reviewer #3: **Yes: **Andrea plutino

---

## [Author Response · Author response to Decision Letter 1]

20 Aug 2024

Reviewer #2: 

1. Comment: There are still some minor problems. The statistician who did the randomization needs to read a book on randomization, perhaps "Randomization in Clinical Trials, Theory and Practice, 2016", to understand the difference between complete randomization and random blocks. Change "complete randomization scheme" to "randomization procedure", because complete randomization is not the same thing as random blocks. For random blocks, you must state the random blocks sizes that were selected. Was the selection of random sizes equally likely? 

Response: Thanks for your kind reminder. We have changed “complete randomization scheme” to “randomization procedure”. Patients were assigned to their treatment arm according to stratified random lists that were balanced in blocks of various sizes in random sequence. To conceal allocation sequence, random permuted blocks with sizes 2 and 4 were used.

1. Comment: In Section 5, just copying my suggestion into the text does not answer the question. WHAT specific assumptions from your pilot of 15 were actually made (e.g., variability, coefficient of variation), and what values were realized in the study? These are simple questions that any biostatistics students would be able to answer. 

Response: Thanks for your kind reminder. In the final paragraph of the Frontiers section, we have outlined the specific assumptions made in our 15-person pilot study. The results of this pilot study can be seen in the sample size calculation section. Additionally, in the Conclusion section, we present the values achieved in the study. Based on the pilot study involving 15 patients, we calculated the mean and standard deviation of the scores for each evaluation index both before and after the intervention. We hypothesized that the three groups would show differential improvements in upper limb motor function, motor control, and joint range of motion, and that the rehabilitation effect of BWS-TCY would surpass that of both RAT and CRT. Furthermore, we posited that the rehabilitation effect of RAT would be greater than that of CRT. The pilot study of 15 patients was generally consistent with our assumptions in the actual study. As clinicians, we greatly appreciate your patient guidance.

Reviewer #3: 

1. Comment: Despite the positive response by the authors my first comment was not addressed:"Introduction

These sentences are repeated in very similar way and are very near; it could be

possible that at least this paragraph was AI generated

128-134 Previous studies have demonstrated that body weight supported Tai Chi gait

training, using a suspension device within a balance bar, can enhance lower limb

motor function and balance in early-stage stroke patients. However, this type of

training typically requires the assistance of two therapists simultaneously, which can

be time-consuming and labor-intensive. Rehabilitation robots, equipped with

exoskeletons and robotic arms, have the capability to provide auxiliary, resistance,

and passive training

142-149 By incorporating Tai Chi with suspension devices, patients who are unable to

fully bear weight can initiate training promptly. However, this training method

requires the presence of at least two therapists simultaneously, resulting in time and

labor consumption. Rehabilitation robots, on the other hand, offer gravity

compensation and are user-friendly[40]. Rehabilitation robotic exoskeletons and

joysticks not only provide gravity compensation but also offer power-assisted training,

esistance training, and passive training.

Response: Thank you for your kind reminder and sorry for our carelessness, we have

removed the duplicate content (Line 128-134)."

Response: Thank you for your kind reminder and sorry for our carelessness, we were using the unmodified version, which has now been corrected.

2. Comment: Furthermore I kindly ask why the founding and the grant numbers were changed from the previous text

Response: Thank you for your concern. Since the fund projects mentioned in the previous text have been closed, we have updated some fund projects that have not yet been closed. So the founding and the grant numbers were changed.

---

## [Decision Letter · Decision Letter 2]

15 Oct 2024

PONE-D-24-10912R2Effects of Body Weight-supported Tai Chi Yunshou training on upper limb motor function in stroke patients: A three-arm parallel randomized controlled trialPLOS ONE

Dear Dr. Huanxia,

Thank you for submitting your manuscript to PLOS ONE. After careful consideration, we feel that it has merit but does not fully meet PLOS ONE’s publication criteria as it currently stands. Therefore, we invite you to submit a revised version of the manuscript that addresses the points raised during the review process.

We look forward to receiving your revised manuscript.

Kind regards,

Domiziano Tarantino, MD

Academic Editor

PLOS ONE

Journal Requirements:

Reviewers' comments:

Reviewer's Responses to Questions

**Comments to the Author**

1. If the authors have adequately addressed your comments raised in a previous round of review and you feel that this manuscript is now acceptable for publication, you may indicate that here to bypass the “Comments to the Author” section, enter your conflict of interest statement in the “Confidential to Editor” section, and submit your "Accept" recommendation.

Reviewer #3: (No Response)

Reviewer #4: (No Response)

2. Is the manuscript technically sound, and do the data support the conclusions?

Reviewer #3: Yes

Reviewer #4: Yes

3. Has the statistical analysis been performed appropriately and rigorously? 

Reviewer #3: Yes

Reviewer #4: Yes

4. Have the authors made all data underlying the findings in their manuscript fully available?

Reviewer #3: Yes

Reviewer #4: Yes

5. Is the manuscript presented in an intelligible fashion and written in standard English?

Reviewer #3: Yes

Reviewer #4: Yes

6. Review Comments to the Author

Reviewer #3: I have to repeat my previous responde because it was not addressed despite your positive response

"Introduction

These sentences are repeated in very similar way and are very near

128-134 Previous studies have demonstrated that body weight supported Tai Chi gait

training, using a suspension device within a balance bar, can enhance lower limb

motor function and balance in early-stage stroke patients. However, this type of

training typically requires the assistance of two therapists simultaneously, which can

be time-consuming and labor-intensive. Rehabilitation robots, equipped with

exoskeletons and robotic arms, have the capability to provide auxiliary, resistance,

and passive training

142-149 By incorporating Tai Chi with suspension devices, patients who are unable to

fully bear weight can initiate training promptly. However, this training method

requires the presence of at least two therapists simultaneously, resulting in time and

labor consumption. Rehabilitation robots, on the other hand, offer gravity

compensation and are user-friendly[40]. Rehabilitation robotic exoskeletons and

joysticks not only provide gravity compensation but also offer power-assisted training,

esistance training, and passive training.

Reviewer #4: Thank you for giving me the opportunity to review this article.

The comments by previous reviewers have been addressed.

I just have one more comment: could the authors please include the effect size and correlation coefficient they used in the sample size calculation?

Please also explain in the article how this effect size was calculated.

Thank you!

7. PLOS authors have the option to publish the peer review history of their article (what does this mean?). If published, this will include your full peer review and any attached files.

Reviewer #3: **Yes: **Andrea Plutino

Reviewer #4: No

---

## [Author Response · Author response to Decision Letter 2]

17 Oct 2024

Reviewer #3: 

1. Comment: Despite the positive response by the authors my first comment was not addressed:"Introduction

These sentences are repeated in very similar way and are very near; it could be

possible that at least this paragraph was AI generated

128-134 Previous studies have demonstrated that body weight supported Tai Chi gait

training, using a suspension device within a balance bar, can enhance lower limb

motor function and balance in early-stage stroke patients. However, this type of

training typically requires the assistance of two therapists simultaneously, which can

be time-consuming and labor-intensive. Rehabilitation robots, equipped with

exoskeletons and robotic arms, have the capability to provide auxiliary, resistance,

and passive training

142-149 By incorporating Tai Chi with suspension devices, patients who are unable to

fully bear weight can initiate training promptly. However, this training method

requires the presence of at least two therapists simultaneously, resulting in time and

labor consumption. Rehabilitation robots, on the other hand, offer gravity

compensation and are user-friendly[40]. Rehabilitation robotic exoskeletons and

joysticks not only provide gravity compensation but also offer power-assisted training,

esistance training, and passive training.

Response: Thank you for your kind reminder and sorry for our carelessness, we have

removed the duplicate content (Line 128-134)."

Response: Thank you for your kind reminder. On August 8, 2024, we have deleted the repeated sentences in the manuscript, and revised it again this time. The modification content is as follows. Previous studies have demonstrated that weight-supported Tai Chi footwork training, which employs a suspension device within a balance bar, enhances lower limb motor function and balance in patients with early-stage stroke[40, 41]. By integrating Tai Chi with suspension devices, patients who are unable to fully bear weight can commence training promptly. However, this training method necessitates the presence of at least two therapists simultaneously, leading to increased time and labor demands. In contrast, rehabilitation robots provide gravity compensation and are more user-friendly[42].

Reviewer #4: 

1. Comment: I just have one more comment: could the authors please include the effect size and correlation coefficient they used in the sample size calculation? Please also explain in the article how this effect size was calculated.

Response: Thank you for your kind reminder. Based on the G*power two-factor repeated measures analysis of variance (ANOVA) F test, the effect size was calculated to be 0.2765, derived from the mean and standard deviation of the FMA-UE scores across the three patient groups. A total sample size of 84 cases was determined for a two-tailed test with a power of 80% , repeat measurement four times, with a correlation among repeated measures of 5%, and a significance level of 5% (alpha error).

---

## [Decision Letter · Decision Letter 3]

5 Nov 2024

Effects of Body Weight-supported Tai Chi Yunshou training on upper limb motor function in stroke patients: A three-arm parallel randomized controlled trial

PONE-D-24-10912R3

Dear Dr. Huanxia,

We’re pleased to inform you that your manuscript has been judged scientifically suitable for publication and will be formally accepted for publication once it meets all outstanding technical requirements.

Kind regards,

Domiziano Tarantino, MD

Academic Editor

PLOS ONE

Additional Editor Comments (optional):

Reviewers' comments:

Reviewer's Responses to Questions

**Comments to the Author**

1. If the authors have adequately addressed your comments raised in a previous round of review and you feel that this manuscript is now acceptable for publication, you may indicate that here to bypass the “Comments to the Author” section, enter your conflict of interest statement in the “Confidential to Editor” section, and submit your "Accept" recommendation.

Reviewer #2: All comments have been addressed

Reviewer #3: All comments have been addressed

Reviewer #4: All comments have been addressed

2. Is the manuscript technically sound, and do the data support the conclusions?

Reviewer #2: (No Response)

Reviewer #3: Yes

Reviewer #4: Yes

3. Has the statistical analysis been performed appropriately and rigorously? 

Reviewer #2: (No Response)

Reviewer #3: Yes

Reviewer #4: Yes

4. Have the authors made all data underlying the findings in their manuscript fully available?

Reviewer #2: (No Response)

Reviewer #3: Yes

Reviewer #4: No

5. Is the manuscript presented in an intelligible fashion and written in standard English?

Reviewer #2: (No Response)

Reviewer #3: Yes

Reviewer #4: Yes

6. Review Comments to the Author

Reviewer #2: (No Response)

Reviewer #3: All comments have been adressed in the last revision so I believe that this article could be published

Reviewer #4: Thank you for addressing my comments.

One last thing, based on your G*Power screenshot, the correlation is 0.5, not 5% as mentioned in your article.

If this paper gets accepted, please correct this before final publication.

7. PLOS authors have the option to publish the peer review history of their article (what does this mean?). If published, this will include your full peer review and any attached files.

Reviewer #2: No

Reviewer #3: **Yes: **Andrea Plutino

Reviewer #4: No

---

## [Editor Report · Acceptance letter]

13 Nov 2024

PONE-D-24-10912R3 

PLOS ONE

Dear Dr. Huanxia, 

I'm pleased to inform you that your manuscript has been deemed suitable for publication in PLOS ONE. Congratulations! Your manuscript is now being handed over to our production team.

Kind regards, 

on behalf of

Dr. Domiziano Tarantino 

Academic Editor

PLOS ONE